# Maternal NAT10 orchestrates oocyte meiotic cell-cycle progression and maturation in mice

Xue Jiang[1,8], Yu Cheng[2,8], Yuzhang Zhu[3,8], Caoling Xu[1,8], Qiaodan Li[4], Xuemei Xing[5], Wenqing Li[1], Jiaqi Zou[1], Lan Meng[1], Muhammad Azhar[1], Yuzhu Cao[5], Xianhong Tong[5], Weibing Qin [6] ✉, Xiaoli Zhu [5] ✉ & Jianqiang Bao [1,7] ✉

In mammals, the production of mature oocytes necessitates rigorous regulation of the discontinuous meiotic cell-cycle progression at both the transcriptional and post-transcriptional levels. However, the factors underlying this sophisticated but explicit process remain largely unclear. Here we characterize the function of N-acetyltransferase 10 (*Nat10*), a writer for N4-acetylcytidine (ac4C) on RNA molecules, in mouse oocyte development. We provide genetic evidence that *Nat10* is essential for oocyte meiotic prophase I progression, oocyte growth and maturation by sculpting the maternal transcriptome through timely degradation of poly(A) tail mRNAs. This is achieved through the ac4C deposition on the key CCR4-NOT complex transcripts. Importantly, we devise a method for examining the poly(A) tail length (PAT), termed Hairpin Adaptor-poly(A) tail length (HA-PAT), which outperforms conventional methods in terms of cost, sensitivity, and efficiency. In summary, these findings provide genetic evidence that unveils the indispensable role of maternal Nat10 in oocyte development.

In mammals, the germ cells are distinct from somatic cells in that they entail two successive cell divisions following one round of DNA replication, producing haploid gametes with parentally exchanged, but equal amounts of genetic DNA transmitted to the offspring[1]. This sexually dimorphic event is achieved through a highly conserved yet tightly regulated process, namely, meiosis. In mice, the primordial germ cells (PGCs) migrate and colonize the genital ridge prior to embryonic day 10.5 (E10.5). Subsequently, stimulated by the microenvironment signaling emanating from the surrounding soma, the PGCs are sexually determined to commit to either male (XY) or female

(XX) germ cells on E11.5. In the female, the ovary is morphologically distinguishable from the male testis by the lack of cord-like structure on E12.5[2]. Unlike the prolonged mitotic arrest of male germ cells in the embryonic testis, female germ cells (oogonia) initiate meiosis quickly starting from E13.5, and sequentially undergo sub-stages of meiotic prophase I (leptotene, zygotene, and pachytene), but are fully arrested at diplotene stage perinatally until pubertal LH signaling[3]. It is well-known that the topoisomerase SPO11-mediated double-strand breaks (DSBs) are a prerequisite for the timely pairing, synapsis, and recombination between homologous chromosomes. Nonetheless, little is

[1]Reproductive and Genetic Hospital, the First Affiliated Hospital of USTC, Division of Life Sciences and Medicine, University of Science and Technology of China (USTC), 230001 Hefei, Anhui, China. [2]School of Information Science and Technology, University of Science and Technology of China (USTC), 230001 Hefei, Anhui, China. [3]Division of Life Sciences and Medicine, University of Science and Technology of China (USTC), 230001 Hefei, Anhui, China. [4]Laboratory animal center, University of Science and Technology of China (USTC), 230001 Hefei, Anhui, China. [5]Reproductive and Genetic Hospital, The First Affiliated Hospital of USTC, Division of Life Sciences and Medicine, University of Science and Technology of China (USTC), 230001 Hefei, Anhui, China. [6]NHC Key Laboratory of Male Reproduction and Genetics, Guangdong Provincial Reproductive Science Institute (Guangdong Provincial Fertility Hospital), 510600 Guangzhou, China. [7]Hefei National Research Center for Physical Sciences at the Microscale, Biomedical Sciences and Health Laboratory of Anhui Province, University of Science and Technology of China (USTC), 230001 Hefei, Anhui, China. [8]These authors contributed equally: Xue Jiang, Yu Cheng, Yuzhang Zhu, Caoling Xu. ✉e-mail: guardqin@163.com; xiaolizh@ustc.edu.cn; jqbao@ustc.edu.cn

known about how genome-wide DSBs are specifically generated and repaired and which factors are involved in this extended stage of meiotic prophase I[4].

After birth, individual oocytes arrested at the diplotene of prophase I are encircled by a flattened layer of granulosa cells, which together constitute the primordial follicles. A clutch of primordial follicles is periodically recruited to the growth phase, undergoing sequentially through stages of primary, secondary, early antral, antral, and large antral follicles, ensued by the ovulation upon LH surge[2]. This process is exquisitely coordinated at the molecular level, as evidenced by massive gene transcription along with selective degradation of substrate transcripts, which together build up the maternal transcriptome. The growing oocytes acquire the meiotic competence, i.e., the capability of resuming meiosis to MII stage, during the transition from the secondary to early antral follicles, while the acquisition of developmental competence occurs in the late stage of antral follicles that gives rise to the matured MII oocytes with the ability to be fertilized and develop to term (referred to as oocyte maturation)[2,5–7]. Since the zygote is transcriptionally silent prior to zygotic gene activation (ZGA), all the initial developmental events during early embryonic reprogramming are dependent on the maternal RNA transcriptome inherited from the oocyte.

Remarkably, the maternal transcriptome is particularly rich in mRNA, which occupies up to 20% of the total RNA in a fully grown GV oocyte, in contrast to an average of ~2% mRNA in somatic cells[8,9]. The transcriptional activity peaks in the early growing oocytes, but decreases thereafter and is considered silent in the fully grown GV oocyte[10]. Noteworthily, this process is concurrent with DNA configuration transition from the less condensed, non-surrounded nucleolus (NSN) to the surrounded nucleolus (SN) state, wherein the fully condensed chromatin DNA encompasses the nucleolus[11–13]. Of note, compared with SN oocytes, NSN oocytes exhibit higher transcriptional activity and lower developmental competence, with most derived zygotes arresting at the 2-cell stage. Moreover, genome-wide transcriptome analyses revealed that SN oocytes display distinct gene expression profiles and metabolic pathway enrichment from NSN oocytes[11,14]. These studies reinforce the significance of maternal transcriptome integrity in coordinating the timely oocyte growth and maturation, and implicate the pathogenic roles, when disrupted, underlying female infertility. Nevertheless, which factors define the maternal transcriptome and how they coordinate with each other through the successive stages of folliculogenesis remain poorly understood.

Our interest in exploring the role of Nat10 in oocyte development was initially piqued for two reasons. First, during oocyte maturation, there is a profound poly(A)-shortening mechanism-mediated mRNA decay that dramatically reshapes the maternal transcriptome. How particular mRNAs are selected for destruction is not well-understood[6,15]. Nonetheless, this event must occur at the post-transcriptional level since the DNA transcription is progressively shut down concurrent with the oocyte growth. Prior studies have shown that multiple RNA modifications, such as m6A, m5C, and m1A, are involved in post-transcriptional RNA metabolism, including alternative splicing, mRNA decay, and mRNA translation[16,17]. Nat10 is highly expressed in the mouse ovary and is by far the only known "writer" for epi-transcriptomic modification- N4-acetylcytidine (ac4C). In HeLa cells, it has been shown to enhance mRNA stability and protein translation[18]. Second, as aforementioned, oocyte development is under spatiotemporally stringent regulation during the discontinuous meiotic cell divisions—rapid progression at early meiotic prophase I in the embryonic gonad, lengthened late prophase I arrest at diplotene during prepubertal development, and quick cytoplasmic and nuclear maturation during the GV-MII transition. Compelling studies have shown evidence related to Nat10's function in cell-cycle control in somatic cells[19–21]. Interestingly, specific deletion of *Nat10* in the male

germline elicited severe defects resulting in male infertility owing to corrupted meiotic cell-cycle progression in mice[22]. This evidence altogether is reminiscent of the critical role of Nat10 in female oocyte development.

In this study, we generate two germline-specific *Nat10* knockout (KO) mouse models by *Stra8*- and *Zp3*-driven Cre expression. We reveal that Nat10 plays a profound role that is indispensable for oocyte meiotic cell-cycle progression. Importantly, we devise a method, termed HA-PAT, that outperforms existing methods adopted for poly(A) tail length examination in single oocytes. Together, we uncover that Nat10-mediated poly(A) tail shortening through maintenance of CCR4-NOT activity is a critical mechanism that defines the oocyte maternal transcriptome, and Nat10 is required for mouse oocyte development by fine-tuning gene expression at both the transcriptional and translational levels.

## Results

### NAT10 is highly expressed and localized to the nucleolus in mouse oocytes

To explore the potential function of Nat10 during oogenesis, we first evaluated the multi-tissue expression pattern of Nat10 in an array of mouse organs. NAT10 protein was abundantly present in the ovary, with the highest levels detected in the thymus, spleen, and testis, which is in support of a recent study (Fig. 1a)[22]. We further re-analyzed the published bulk RNA-seq datasets in the growing oocytes and pre-implantation embryos (GSE71434)[23]. It showed that Nat10 mRNA was dynamically regulated in the postnatal growing follicles and pre-implantation embryos, with the highest mRNA levels detectable in GV oocytes and the lowest levels in 2-cell embryos (Fig. 1b). In agreement with this finding, quantitative real-time PCR (qPCR) validated the similar expression trend of Nat10 mRNA levels spanning different stages of oocytes and embryos (Fig. 1c). Next, we performed the fluorescent immunostaining (IF) with a NAT10 antibody in isolated oocytes at various stages as well as in ovary cryosections. As shown in Fig. 1d, the NAT10 protein is abundantly present in the nucleus of the GV oocytes. The intensity of the NAT10 signal is reduced and dispersed in the nucleus of MI and MII oocytes owing to the breakdown of the nuclear membrane (Fig. 1d). Based on the morphology and the number of surrounding granulosa cells, the follicles can be categorized into distinct stages during postnatal folliculogenesis in mice, including primordial follicle (PrF), the primary follicle (PF), the secondary follicle (SF), the early antral follicle (EAF), and antral follicle (AF) (Fig. 1e)[24].

Consistent with its mRNA expression trend, IF revealed abundant NAT10 protein being detected in the central nucleus of the growing oocytes at various stages, with the highest intensity in the nucleus center encircled by a layer of highly condensed chromatin (Fig. 1e). This distinctive expression pattern raised the possibility that it might be localized in the nucleolus, as reported by a few previous studies[21,25,26]. To verify this possibility, we co-immunostained NAT10 and the nucleolus-specific marker, nucleophosmin (NPM). As shown in Fig. 1f, NAT10 is well co-localized to the nucleolus with NPM in the oocytes across various stages of folliculogenesis. It is worth noting that NAT10 is also highly expressed in the nucleolus of granulosa cells, particularly in antral follicles, as revealed by its perfect co-localization with NPM (Fig. 1f). Furthermore, we tracked down the expression dynamics of NAT10 protein from growing oocytes (GO) to pre-implantation embryos. This revealed that NAT10 protein diffuses from the nucleolus to the nucleoplasm in NSN and SN oocytes, and fills in the whole cytoplasm in the MI and MII oocytes (Supplementary Fig. 1a). Its expression levels, however, are reduced in 2-cell and 4-cell embryos, but abundantly present in the nucleolus in 8-cell, morula, and blastocyst. Of note, the NAT10 signal is a bit enriched in the spindles of MI and MII oocytes owing to the nuclear breakdown (Supplementary Fig. 1a). This evidence altogether implies that NAT10 likely has important physiological roles through mouse oocyte development in vivo.

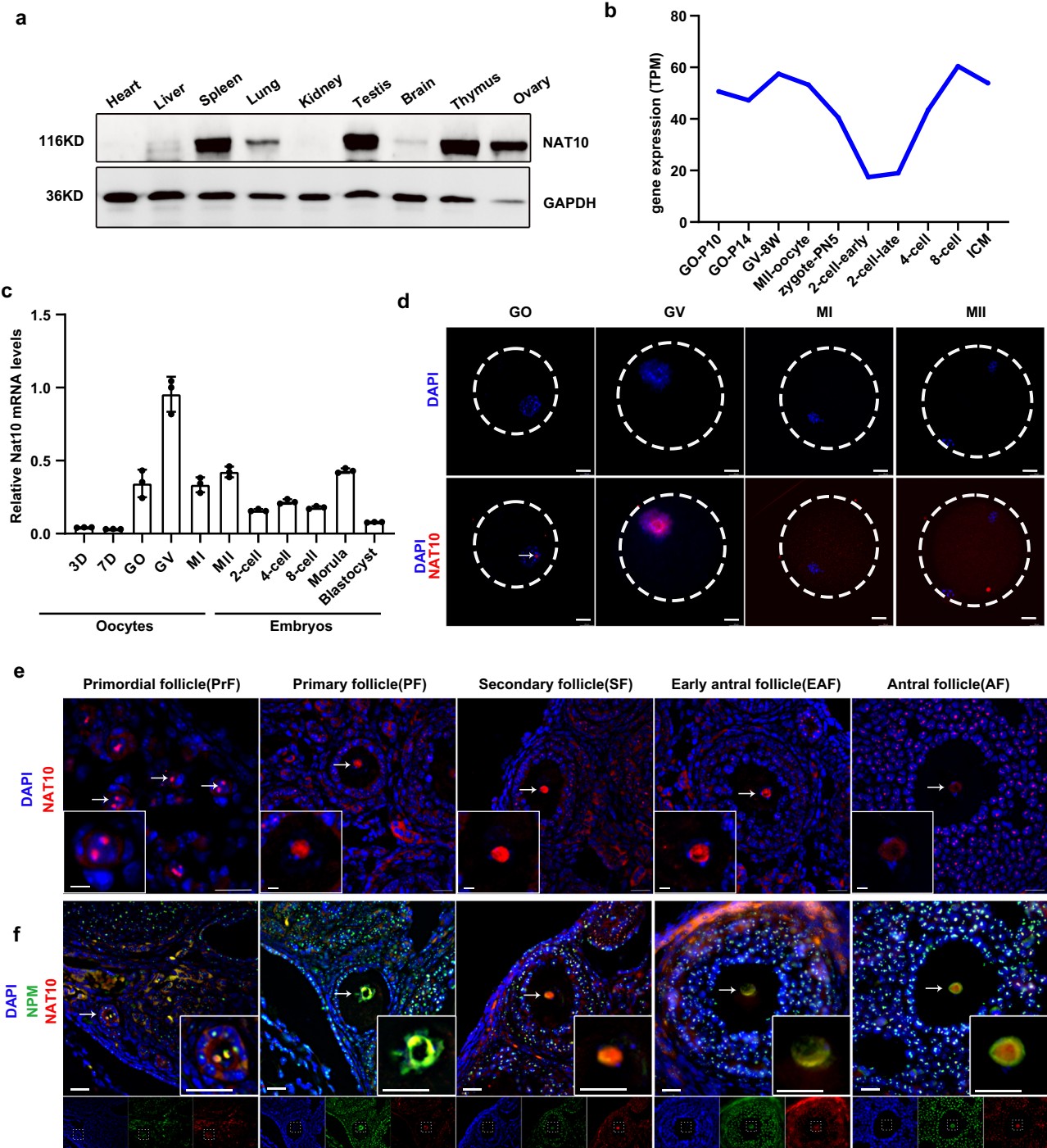

**Fig. 1 | Expression and Localization of Nat10 enriched in the nucleolus of oocytes in mice. a** Western blot showing the relative expression levels of NAT10 protein among multiple organs in adult WT mice. GAPDH served as a loading control. *n* = 3 biologically independent samples were included in each group. **b** Dynamic mRNA expression levels of Nat10 from RNA-seq analyses in oocytes and preimplantation embryos in mice (GSE71434). ICM, Inner cell mass. **c** Quantitative RT-PCR results showing the relative expression levels of mouse Nat10 mRNA in oocytes, and preimplantation embryos. Data were presented as mean ± SEM, *n* = 3. GO, growing oocytes collected from postnatal 14-day-old (P14) female mice. **d** Immunofluorescence (IF) staining of NAT10 in growing (GO), GV, MI, and MII oocytes as indicated. Dashed circle indicates cellular membrane of oocytes. DNA was counterstained with 4′,6-diamidino-2-phenylindole (DAPI). Scale bar, 20 μm. **e** IF images of 21-day-old WT ovarian cryosections stained with anti-NAT10 antibody

(Red) and DAPI (Blue) for follicles at various stages (primordial, primary, secondary, early antral, and antral stages) as indicated. Scale bar, 20 μm. *n* = 3 biologically independent samples were included in each group. Boxed inset area is a magnified view of the oocyte in the respective follicles. Arrows point to NAT10-positive nucleus of the oocyte in mouse-developing follicles. **f** IF images of 21-day-old WT ovarian sections co-stained with NAT10 antibody (Red), Nucleophosmin (NPM, Green), and DAPI (Blue) in the follicles as indicated. Scale bar, 20 μm. *n* = 3 biologically independent samples were included in each group. Bottom panel is a magnified view of the oocyte in the respective follicles. Arrows point to the co-localization of NAT10 and NPM in the oocyte nucleolus at varied stages of follicles. *n* = 3 biologically independent samples were included in each group (**a**–**c**). Source data are provided as a source data file.

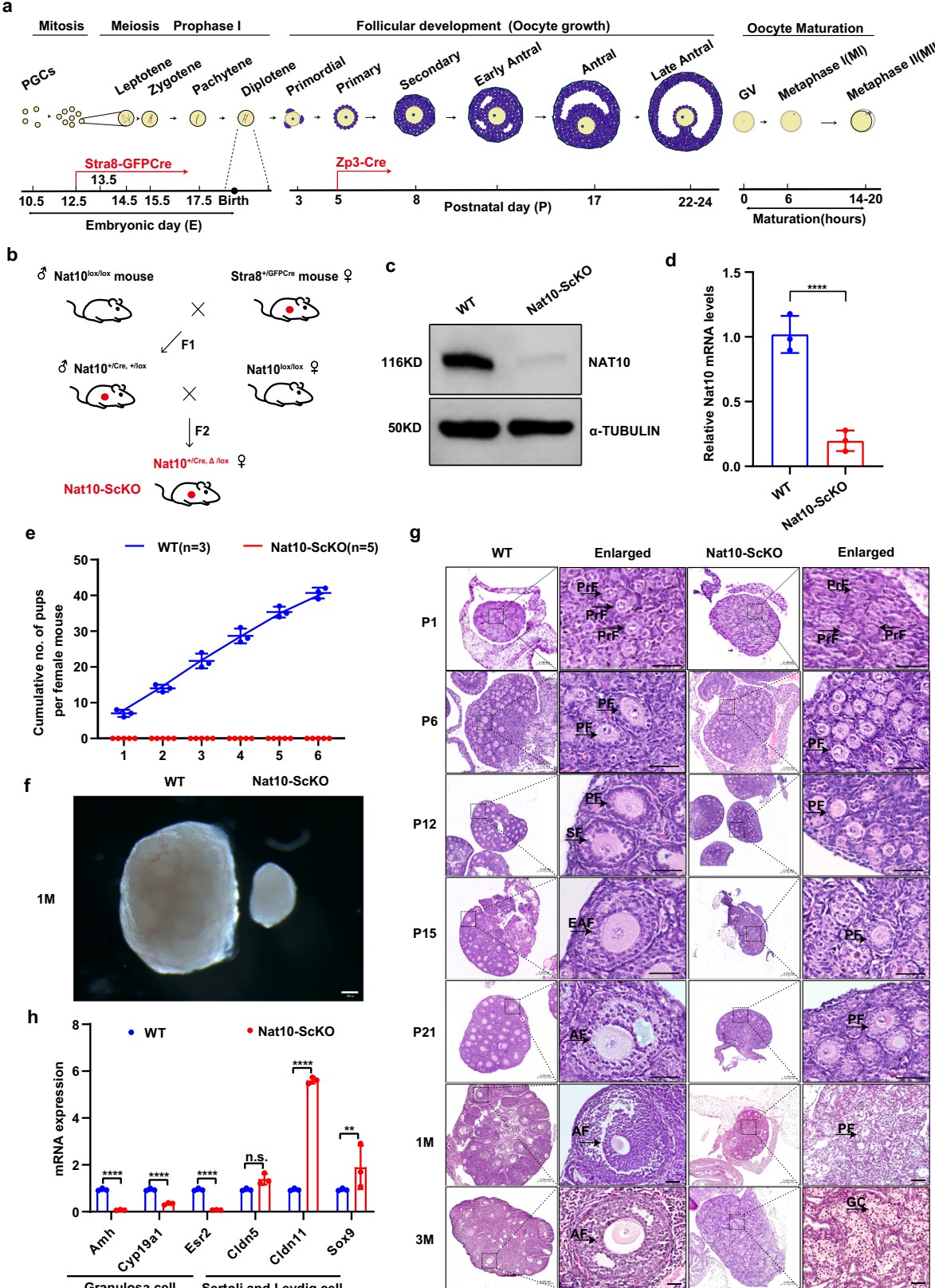

## Pre-meiotic deletion of *Nat10* caused follicular arrest at primary follicles and premature ovarian failure (POF)

Next, to decipher the physiological function of Nat10 in vivo, we generated pre-meiotic stage-specific *Nat10* knockout (KO) mice by crossing floxed Nat10 (Nat10^lox/lox) alleles with Stra8-GFPCre knockin alleles to obtain oocyte conditional Nat10 KO females (Nat10^lox/-; Stra8-

GFPCre, hereafter called Nat10-ScKO) (Supplementary Fig. 1b–d)[27]. The floxed Exons #4 and #5 (E4/5) reside in the DUF1726 domain of NAT10 protein (Supplementary Fig. 1b). Stra8-GFPCre is specifically activated around embryonic day 12 (E12) in the primordial germ cells (PGCs) prior to meiosis in the female embryonic gonad (Fig. 2a)[28]. When crossing these deletor mice with *Nat10^lox/lox* mice, E4/5 of the *Nat10*

**Fig. 2 | Pre-meiotic deletion of Nat10 caused follicular developmental arrest and premature ovarian failure (POF). a** Schematic diagram showing the landmark timeline of oocyte development from embryonic meiotic cell-cycle progression to postnatal oocyte growth and maturation. Stra8-GFPcre is activated prior to Embryonic day 13.5 (E13.5); Zp3-cre is active starting from P5 in the primary follicles. **b** A breeding scheme by crossing Nat10[lox/lox] with Stra8-GFPCre to generate Nat10[lox/-]; Stra8-GFPCre (Nat10-ScKO) offspring. **c** Western blot analyses of the NAT10 protein levels in adult WT and Nat10-ScKO ovary. α-TUBULIN was used as a loading control. **d** Quantitative RT–PCR (qPCR) assay showing the relative expression levels of Nat10 mRNA in adult WT and Nat10-ScKO mouse ovary. Data are presented as mean ± SEM, $n = 3$. ****$p < 0.0001$ by two-tailed *Student's t-test*. **e** Fertility test showing the cumulative average numbers of pups from breeding of WT and Nat10-ScKO females with WT males. Data are presented as the mean ± SEM, $n = 5$, ****$p < 0.0001$ by two-tailed *Student's t-test*. **f** The gross morphology of ovaries derived from WT and Nat10-ScKO mice at 1 M. Scale bar, 200 μm. $n = 3$ biologically independent samples were included in each group. **g** H&E staining of paraffin-embedded ovarian sections showing the histology of WT and Nat10-ScKO ovaries at postnatal days as indicated. Scale bar, 200 μm. A high-resolution view of the boxed area is shown in parallel. Scale bar, 20 μm. $n = 3$ biologically independent samples were included in each group. Arrows point to follicles at stages as indicated. PrF primordial follicle, PF primary follicle, SF secondary follicle; EAF early antral follicle, AF antral follicle, LAF late antral follicle. **h** qPCR analyses of the relative expression levels for a cohort of genes showing specific or characteristic expression in ovarian granulosa cells (Left) or testicular Sertoli/Leydig cells (Right) in 1-month-old WT and Nat10-ScKO ovaries. Data are presented as the mean ± SEM, $n = 3$. n.s., not significant; **$p < 0.01$; ****$p < 0.0001$ by two-tailed *Student's t-test*. Cldn5, $p = 0.3326$; Sox9, $p = 0.0059$. $n = 3$ biologically independent samples were included in each group (**c, f, g**). Source data are provided as a source data file.

gene was removed, resulting in the frame-shift translation and presumably nonsense-mediated mRNA decay (NMD) (Fig. 2a–d, Supplementary Fig. 1b, c)[27]. Using isolated GV oocytes, both western blotting and qPCR verified the markedly reduced protein and mRNA levels of Nat10 in Nat10-ScKO oocytes compared with WT oocytes (Fig. 2c, d). Fertility testing showed that female Nat10-ScKO mice were completely sterile when crossed with WT males during 6 months of breeding (Fig. 2e). At 1 month, the Nat10-ScKO ovary was reduced by eightfold in size compared with the WT ovary (Fig. 2f).

To trace at which stage the *Nat10* abrogation impacted follicle development, we next examined the histology of Hematoxylin&Eosin (HE)-stained sections using paraffin-embedded ovaries during the first wave of postnatal follicle development. This unveiled that the follicles managed to proceed but arrested at the stage of primary follicles before P21 in the Nat10-ScKO females, unlike the WT ovaries wherein antral follicles were frequently observed (Fig. 2g). After P21, the Nat10-ScKO ovaries progressively lost all the characteristic follicular structure but harbored tubule-like structures filled up with homogeneous, immature granulosa cells, resembling the seminiferous tubules in the male testis (Fig. 2g). These tubules were devoid of oocytes in the center, presumably owing to the quick degeneration of the *Nat10-null* oocytes, reminiscent of premature ovarian failure (POF) (Fig. 2g). Indeed, this phenomenon for the transdifferentiation of ovarian cells to Sertoli-like cells has been previously reported, wherein deletion of *Mtor* in the primordial oocytes induced the conversion of granulosa cells to Sertoli-like cells, displaying a seminiferous tubule-like testicular structure in the oocyte-specific *Mtor-null* ovaries[29]. To test this possibility, we next designed a panel of granulosa cell- and Sertoli/Leydig cell-specific primers. As shown in Fig. 2h, while the expression levels of granulosa cell-specific markers (*Amh*, *Cyp19a1,* and *Esr2*) were significantly reduced in Nat10-ScKO ovaries compared with WT ovaries, we only observed the mRNA expression levels of two markers, *Sox9* in Sertoli cells and *Cldn11* in Leydig cells, among a selection of testis-enriched markers (*Cyp11b1*, *Hsd3b6*, *Gata1*) were markedly elevated. This evidence implicated the partial conversion of granulosa cells to Sertoli cells upon *Nat10* KO. In summary, pre-meiotic ablation of *Nat10* in the female gonad led to female infertility due to follicular arrest at the primary follicle stage and premature ovarian failure.

### Pre-meiotic loss of *Nat10* led to oocyte meiotic arrest at pachytene stage

Unlike the male germline, the female germ cells initiate meiotic prophase I division early following sex determination in the embryonic gonad, and sequentially undergo leptotene, zygotene, and pachytene, but are fully arrested at the diplotene stage perinatally (Fig. 2a). During the preparation of this manuscript, an essential role for Nat10 in meiotic divisions was reported in the testis[22], thus we next investigated whether Nat10 is required for female meiotic prophase I progression in vivo. Co-staining of the nuclear chromosome spreads by SYCP1 and SYCP3 markers discovered the accumulation of aberrant pachytene-like cells with partially synapsed homologous chromosomes in the perinatal Nat10-ScKO ovaries (Fig. 3a, Supplementary Fig. 1e). Statistical comparisons disclosed the elevated percentage of pachytene/pachytene-like cells and, as a result, the decreased number of oocytes at the diplotene stage in Nat10-ScKO ovaries (Fig. 3a, b), suggesting the meiotic arrest of Nat10-ScKO oocytes at the pachytene stage.

Premature oocyte death during the first wave of folliculogenesis, as described above (Fig. 2g), is most often a consequence of a self-surveillance mechanism for the host to safeguard genome integrity against unsynapsed chromosomes or excessive double-strand DNA breaks (DSBs)[30–32]. A much higher occurrence of aberrant chromosomes not fully synapsed, was observed in the perinatal Nat10-ScKO ovaries (Fig. 3a, b), we thus next assessed the causative factors that account for the premature oocyte loss upon *Nat10* KO. IF staining by γH2AX revealed the elevated DSB signals in oocytes at the pachytene stage, but not at the diplotene stage, in embryonic Nat10-ScKO ovaries (Fig. 3c, d), suggesting defective DSB repair in Nat10-null pachytene oocytes[33]. Further examination by staining with RPA2, a marker that exclusively labels unrepaired DSBs, unveiled that more RPA2 foci were present in the Nat10-ScKO oocytes at pachytene stage, rather than at diplotene stage (Fig. 3e, f), which presumably point to oocytes with elevated γH2AX staining resulting from DSB repair deficiency in Nat10-ScKO ovaries[33]. Together, this evidence suggests that Nat10 is essential for meiotic prophase I progression in the female embryonic gonad.

### Nat10 is essential for the chromatin NSN–SN transition in the growing oocytes

At birth, the oocytes arrest at the diplotene stage in meiotic prophase I, and each is surrounded by a single layer of flattened granulosa cells, which together constitute the primordial follicle pool (Fig. 2a). Upon pubertal stimulation by FSH and LH, the follicles sequentially enter the growth and maturation stages (Fig. 2a). We thus next evaluated whether Nat10 is required for oocyte growth through generation of a *Nat10*-specific deletion mouse model in growing oocytes by crossing the Nat10[lox/lox] alleles with female *Zp3-Cre* (henceforth termed Nat10-ZcKO), which is specifically activated in the oocytes of primary follicles (Figs. 2a and 4a)[27,34]. IF staining using P21 ovaries and isolated GV oocytes showed that NAT10 protein is specifically eliminated from the oocytes, but not in the granulosa cells (Fig. 4b, c, Supplementary Fig. 2a, b). During a half-year fertility testing, Nat10-ZcKO females were completely sterile, suggesting *Nat10* is indispensable for oocyte growth during postnatal ovarian development (Fig. 4d, e). We next performed H&E staining and counted the average number of follicles at various stages in the postnatal ovary sections. At 1 month, there was a slight reduction in the size of the Nat10-ZcKO ovary (Fig. 4e), but the morphological features and the proportion of follicles at various stages were indistinguishable between the Nat10-ZcKO and WT ovaries (Fig. 4e–g). Nevertheless, after the first wave of folliculogenesis, Nat10-ZcKO oocytes appeared to quickly degenerate, resulting in

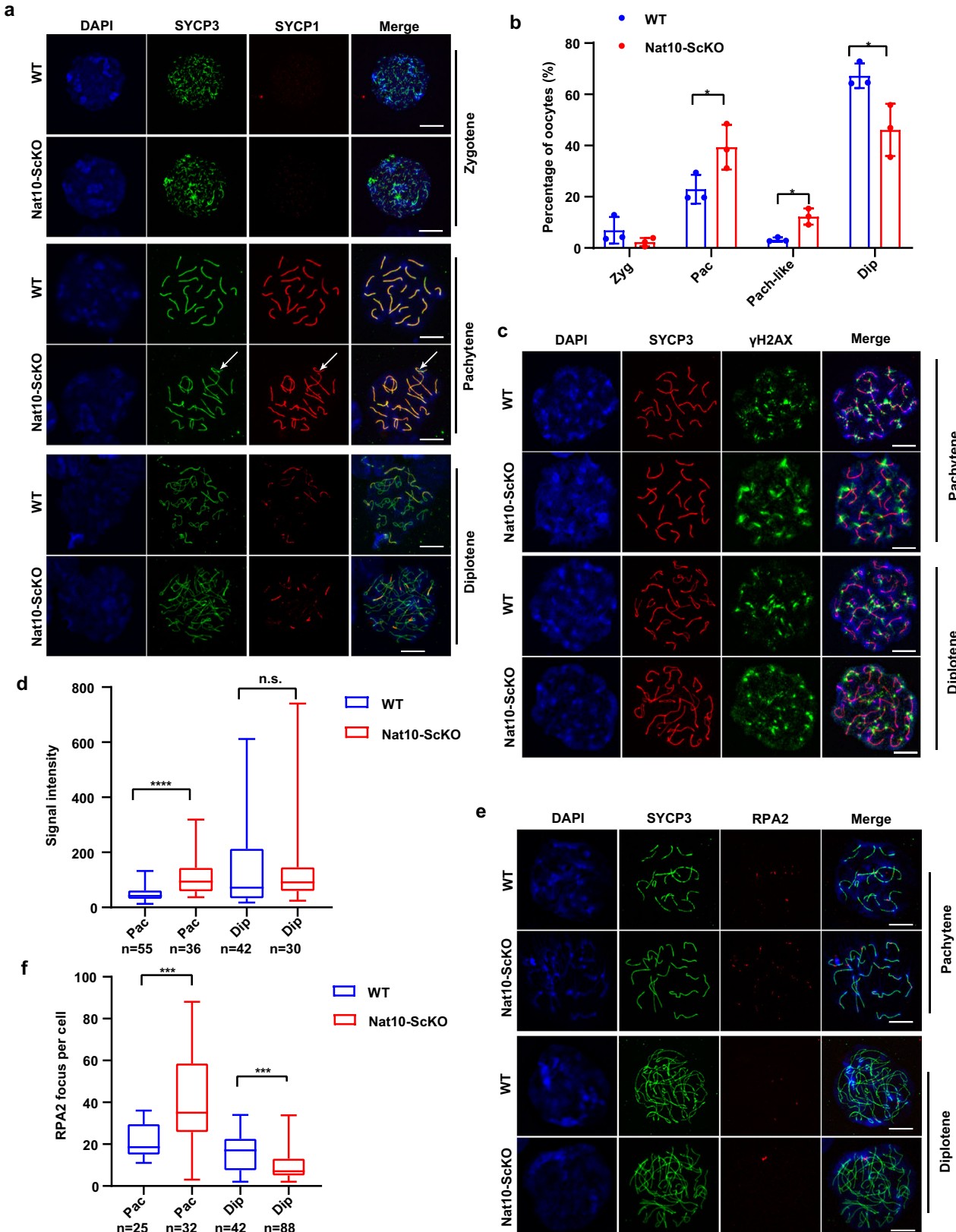

developmental arrest at secondary follicles in Nat10-ZcKO ovaries, as compared with WT ovaries (Fig. 4e–g, Supplementary Fig. 2c, d).

On the other hand, the permissive sterile phenotype in all the Nat10-ZcKO females implies that there exists a functional deficiency in Nat10-ZcKO oocytes despite their morphological similarity in WT ovaries at the age of 1 month. Therefore, we collected and stained

oocytes from the PMSG-primed females at P21. The average numbers of GV oocytes retrieved were comparable between Nat10-ZcKO and WT ovaries (Fig. 5a, b). Notably, more bulged granules were observed in the cytoplasm of Nat10-ZcKO oocytes (Fig. 5a). In growing oocytes, cytoplasmic maturation is accompanied by the non-surrounded nucleolus (NSN)-to-surrounded nucleolus (SN) DNA configuration

**Fig. 3 | Embryonic Nat10 loss caused oocyte meiotic prophase I arrest at pachytene stage owing to deficient DSB repair. a** Immunofluorescence staining of oocyte nuclear chromosome spreads by SYCP3 and SYCP1 markers in WT and Nat10-ScKO mouse ovaries at birth. Scale bar, 10 μm. Arrows point to the asynapsed structure of the lateral and central axes. **b** The statistical counts showing the percentage of oocytes at various stages as indicated. Data are presented as the mean ± SEM, $n = 3$, *$p < 0.05$ by two-tailed *Student's t-test*. Pac, $p = 0.0209$; Pach-like, $p = 0.0387$; Dip, $p = 0.0315$. **c** IF staining by SYCP3 and γH2AX on surface-spread oocytes at pachytene and diplotene stages from WT and Nat10-ScKO mouse oocytes at birth. Scale bar, 10 μm. **d** The statistic counts showing the relative γH2AX signal intensity calculated by ImageJ in pachytene and diplotene oocytes. The box indicates the upper and lower quantiles, the thick line in the box indicates the median and whiskers indicate 2.5th and 97.5th percentiles. Data are presented as mean ± SEM, $n = 3$; ****$p < 0.0001$; n.s., not significant by two-tailed *Student's t-test*. Dip, $p = 0.9182$. **e** IF staining on surface-spread oocytes by SYCP3 and RPA2 in WT and Nat10-ScKO mouse oocytes at birth. Scale bar, 10 μm. **f** Quantification of the numbers of RPA2 foci (representative of the unrepaired DSBs) in WT and Nat10-ScKO mouse oocytes at birth. Zyg zygotene, Pac pachytene, Pac-like pachytene-like, Dip diplotene. The box indicates the upper and lower quantiles, the thick line in the box indicates the median and whiskers indicate 2.5th and 97.5th percentiles. Data are presented as the mean ± SEM, $n = 3$; ***$p < 0.001$ by two-tailed *Student's t-test*. Pac, $p = 0.0001$; Dip, $p = 0.0002$. $n = 3$ biologically independent samples were included in each group (**a, c, e**). Source data are provided as a source data file.

transition[11]. SN oocytes mostly dominate the GV stage in late antral follicles and are considered meiotically competent. Compelling studies have previously shown that DNA is transcriptionally inert in SN oocytes, characterized by the enhanced modifications of H3K4me3 and H3K9me3, while NSN oocytes exhibit active DNA transcription with decreased H3K4me3 and H3K9me3 decoration[14]. Indeed, further counting of NSN versus SN oocytes uncovered that the ratio of NSN to SN was distorted between Nat10-ZcKO and WT ovaries (Fig. 5c, d). H3K4me3 staining showed markedly decreased H3K4me3 intensity in both NSN and SN oocytes in Nat10-ZcKO oocytes compared with WT oocytes (Fig. 5e, f). In contrast, H3K9me3 staining revealed dispersed and elevated chromatin signals in both NSN and SN oocytes in Nat10-ZcKO oocytes (Fig. 5g, h). These results suggest that Nat10 loss altered the chromatin environment for the transcriptional machinery resulting in the compromised meiotic competence in Nat10-ZcKO oocytes. It has been documented that the dys-regulated chromatin signature associates closely with defective NSN to SN transition as evidenced in *Zfp36l2*- and *Rps26*-null oocytes, which presumably resulted from the insufficient expression of two key paracrine growth factors Gdf9 and Bmp15[14,35]. Together, these studies demonstrate that Nat10 is required for oocyte growth in developing follicles, and hence for female fertility.

### *Nat10* ablation impaired meiotic GV-MII progression

The sterility and premature oocyte death in the Nat10-ZcKO females suggested that Nat10 is pivotal for meiotic maturation (GV-MII transition). To test this possibility, we next sought to isolate and culture GV oocytes in vitro and determine whether Nat10-ZcKO oocytes are capable of resuming meiosis. At 3 h after IBMX release in M16 medium, most WT GV oocytes (~90%) resumed meiotic division and entered the pro-metaphase I, as evidenced by the nuclear membrane breakdown of the germinal vesicles (GVBD), while a lower percentage of Nat10-ZcKO GV oocytes (~58%) managed to complete GVBD (Fig. 6a, b, Supplementary Fig. 4a–d). Consistently, a smaller number of Nat10-ZcKO oocytes progressed to the MII stage compared to that of WT oocytes ($23 ± 2.5$ vs. $65 ± 1.51$) (Fig. 6c). To determine the accurate stage and how meiotic divisions were impacted in Nat10-ZcKO oocytes, we collected the superovulated oocytes in vivo after PMSG/hCG injection and performed the co-staining with DAPI and a cytoskeleton marker, tubulin, in the formaldehyde-fixed oocytes. The average number of collected oocytes significantly declined in Nat10-ZcKO ovaries compared with WT ovaries ($3.8 ± 1.15$ vs. $30.57 ± 0.92$) (Fig. 6d). In agreement with previous findings, most Nat10-ZcKO oocytes were arrested at the MI stage, with a small fraction of oocytes exhibiting anaphase-to-telophase arrest in Prophase I (AI-TI) in Nat10-ZcKO ovaries compared to WT oocytes (Fig. 6e, f). Furthermore, we carried out an In vitro fertilization (IVF) assay using superovulated MII oocytes. Nat10-ZcKO oocytes appeared to be fertilized and developed to the 2-cell stage, like the WT oocytes (Fig. 6g, h). However, the proportion of four-cell stage embryos markedly declined in the *Nat10* KO group. Together, these studies suggest that Nat10 is required for oocyte meiotic maturation (Fig. 6).

Oocyte growth is under stringent transcriptional regulation, with peak transcription occurring in growing oocytes at the early stage of antral follicles[6,14,36]. To dissect the factors underlying defective GV oocytes in Nat10-ZcKO ovaries, we carried out RNA-seq analyses with GV oocytes retrieved from WT and Nat10-ZcKO ovaries. In agreement with the sterile severity, a total of 1615 differentially expressed genes (DEGs) were identified (Cutoff: Fold change (FC) ≥ 2, $p < 0.05$), with similar numbers of genes up-regulated (839) and down-regulated (776) in the Nat10-ZcKO GV oocytes relative to the WT oocytes (Supplementary Fig. 3a, c, Supplementary Data 4). Interestingly, Gene ontology (GO) analyses revealed that most down-regulated genes were enriched in transcription-related pathways, whereas the up-regulated genes were related to tRNA processing and meiotic cell cycles (Supplementary Fig. 3b, d). Moreover, there were a total of 583 genes related to cell-cycle progression showing an alternative splicing pattern in Nat10-ZcKO oocytes (Supplementary Fig. 3e–h). This evidence altogether demonstrated that the maternal transcriptome was disrupted in Nat10-ZcKO oocytes at the GV stage responsible for the impaired oocyte maturation.

### Mini-bulk SMART-seq2 identified defective maternal mRNA decay upon Nat10 loss

Oocyte meiotic progression necessitates tightly regulated RNA transcription and degradation[6,37–40]. Transcription is active in growing oocytes but shut down progressively during the oocyte transition from the NSN- to the SN-type GV stage[41]. Upon meiosis resumes, the GV maternal transcriptome undergoes a global but selective degradation of ~20% poly(A) mRNAs, culminating in a characteristic maternal transcriptome in MII oocytes that substantially differs from that in GV oocytes[7,15,42,43]. To interrogate the molecular mechanism underlying the disrupted oocyte maturation in Nat10-ZcKO oocytes, we performed RNA-seq analyses. Given that the Nat10-ZcKO female only superovulated ~4 MII oocytes on average, we further optimized an *in-house* mini-bulk SMART-seq2 protocol that utilized 3–5 oocytes for each biological replicate for RNA-seq (Fig. 7a, b)[27]. We verified the validity of our mini-bulk SMART-seq2 method by comparing our data with published bulk RNA-seq results in WT oocytes. On average, our method detected ~13,337 genes in GV and ~12,071 genes in MII, which are comparable to the ~13,629 genes and ~12,045 genes detected in GV and MII oocytes (Cutoff: TPM ≥ 1), respectively, in the bulk oocyte RNA-seq datasets (Fig. 7b, Supplementary Data 4 and 5)[23].

Next, we conducted the mini-bulk SMART-seq2 using 5 oocytes at MII stage from WT and Nat10-ZcKO females. This revealed a higher number of genes (1196) up-regulated than down-regulated (555) in Nat10-ZcKO MII oocytes (Fig. 7c, Supplementary Fig. 4e, f, Supplementary Data 5). GO analyses showed that the up-regulated genes were mostly enriched in translation- and mRNA-processing-related biological processes (Fig. 7d–f). By comparison, the down-regulated genes were enriched in transcription-related GO terms (Supplementary Fig. 4g). A further comparison showed that a higher number of transcripts was present in the Nat10-ZcKO MII oocytes than in the WT (TPM ≥ 1) (Fig. 7g, h), suggesting an aberrant accumulation of maternal

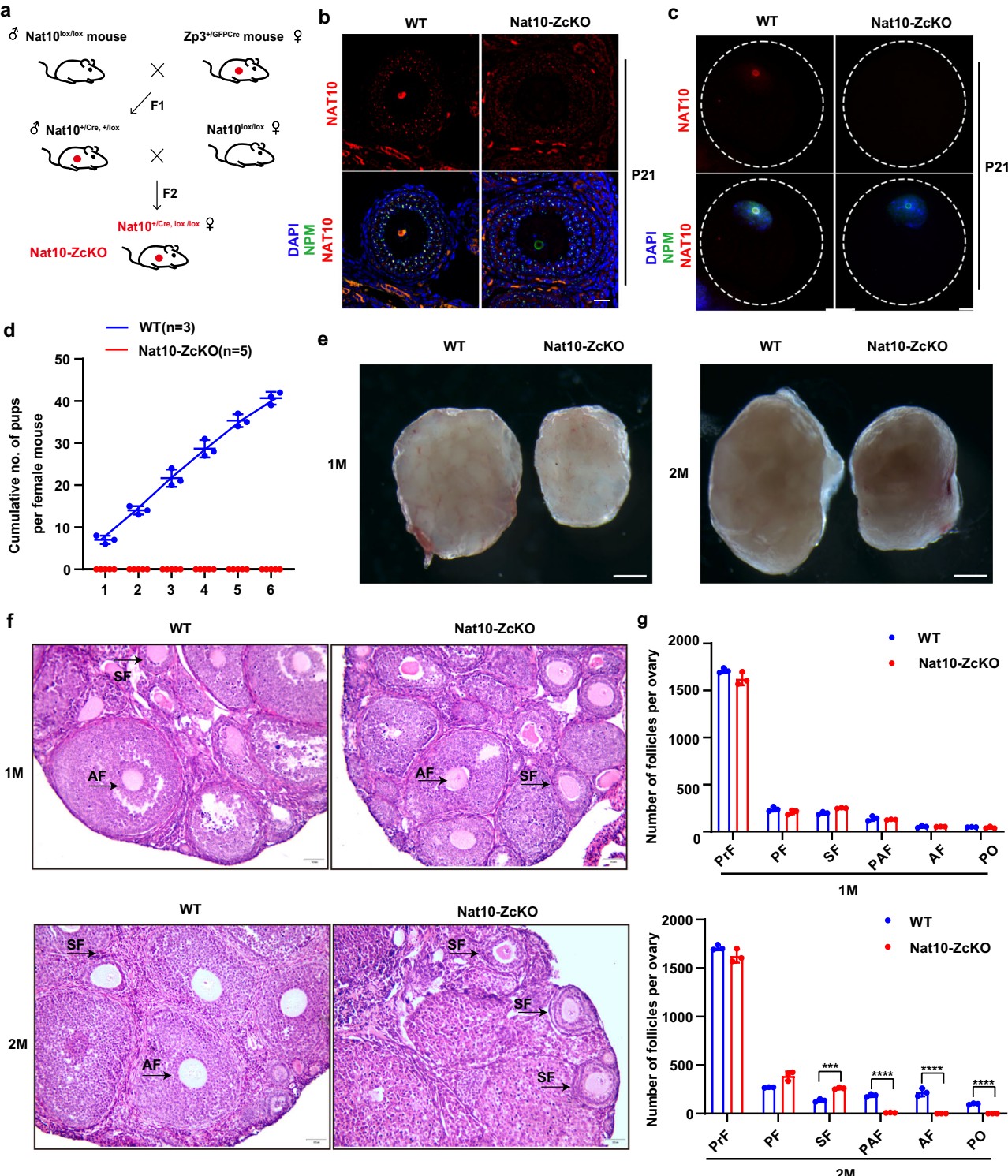

**Fig. 4 | Postnatal Nat10 depletion caused ovarian developmental arrest at secondary follicles. a** A breeding scheme for Nat10 KO in growing oocytes of primary follicles by crossing Nat10$^{lox/lox}$ with Zp3-Cre deleter to attain Nat10$^{lox/-}$; Zp3-Cre (Nat10-ZcKO) female offspring. **b, c** Immunofluorescence staining by NAT10 (Red), NPM (Green), and Hoechst 33342(Blue) in the secondary follicles in **b** and GV oocytes in **c** from WT and Nat10-ZcKO ovaries. Scale bar, 20 μm. **d** Fertility test showing the cumulative numbers of pups from breedings of WT and Nat10-ZcKO females with WT males during a half-year caging. Data are presented as the mean ± SEM, $n = 5$; ****$p < 0.0001$ by two-tailed *Student's t-test*. **e** The gross morphology of ovaries derived from WT and Nat10-ZcKO mice at the age of 1 month (M) (Left) and 2 months (Right). Scale bar, 200 μm. **f** H&E staining showing ovarian histology from WT and Nat10-ZcKO mice at 1M (Top) and 2 M (Bottom). Scale bar, 50 μm. Follicles are indicated by arrows. **g** Comparison of the average numbers of follicles at indicated stages in the ovaries of WT and Nat10-ZcKO mice at 1M (Top) and 2M (Bottom). Follicles were counted on serial ovarian sections after H&E staining. Data are presented as the mean ± SEM, $n = 3$; ***$p < 0.001$; ****$p < 0.0001$ by two-tailed *Student's t-test*. 2 M SF, $p = 0.0003$. PO, Preovulatory Follicle. $n = 3$ biologically independent samples were included in each group (**b, c, e, f**). Source data are provided as a source data file.

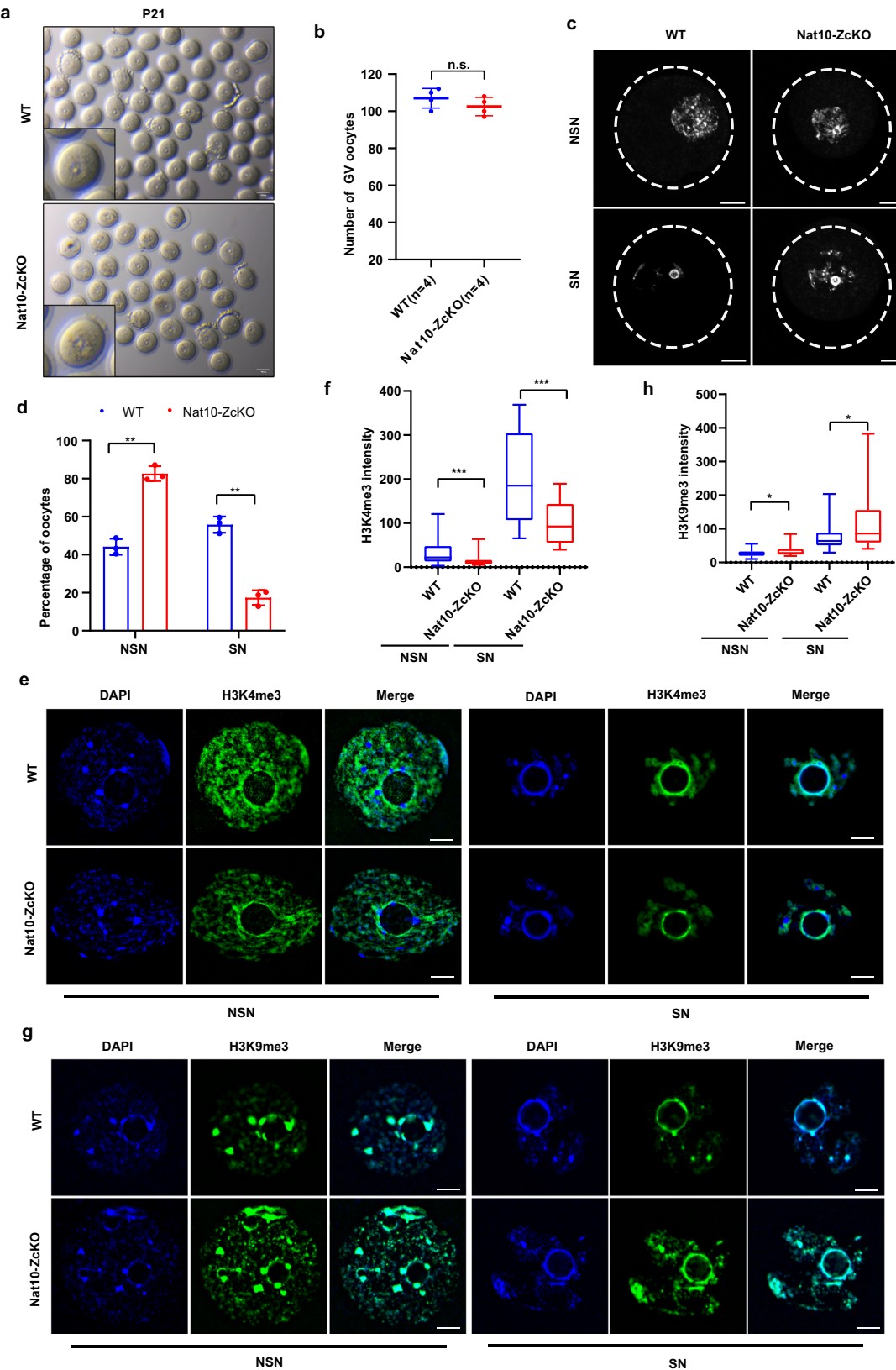

transcripts. To distinguish what specific types of transcripts were affected, the expressed transcripts were divided into five bins according to their expression levels in the WT MII oocytes. This revealed that *Nat10* KO caused global transcript up-regulation regardless of their expression abundance (Fig. 7i). Since there exists global poly(A) mRNA degradation during GV-MII transition in WT

oocytes, the expressed transcripts in WT MII oocytes were allocated into three types: Up- [FC(MII/GV) ≥ 2, $p < 0.05$], Down- [FC(MII/GV) ≤ −2, $p < 0.05$], and Stable-type (remaining transcripts). Sankey plot showed that most up-regulated genes (965/1196) in the Nat10-ZcKO MII oocytes overlapped with the Down-type genes in the WT MII oocytes, suggesting that the 965 transcripts destined for decay during

**Fig. 5 | Postnatal Nat10 deficiency impedes oocyte chromatin NSN–SN configuration transition. a** The gross morphology of oocytes at Germinal vesicle (GV) stage collected from PMSG-primed WT and Nat10-ZcKO females at P21. Scale bar, 100 μm. **b** Quantification of collected average numbers of GV oocytes. Data are presented as the mean ± SEM, $n = 4$. n.s., non-significant by two-tailed *Student's t-test*. **c** Hoechst 33342 (Blue) staining of the GV oocytes with non-surrounded nucleolus (NSN) and surrounded nucleolus (SN) chromatin configurations in WT and Nat10-ZcKO oocytes. Scale bar, 20 μm. **d** The percentage of NSN-type and SN-type oocytes isolated from WT and Nat10-ZcKO mice at P21. Data are presented as the mean ± SEM, $n = 3$. **$p < 0.01$ by two-tailed *Student's t-test*. NSN, $p = 0.003$; SN, $p = 0.003$. **e, f** Immunofluorescence staining by H3K4me3 in NSN-type (Left) and SN-type (Right) oocytes from PMSG-primed WT and Nat10-ZcKO mice in **e**, and quantification of H3K4me3 intensity in (**f**). Scale bar, 10 μm. The box indicates the upper and lower quantiles, the thick line in the box indicates the median and whiskers indicate 2.5th and 97.5th percentiles. Data are presented as the mean ± SEM, $n = 3$; ***$p < 0.001$ by two-tailed *Student's t-test*. NSN, $p = 0.0006$; SN, $p = 0.0003$. **g, h** Immunofluorescence staining by H3K9me3 in NSN-type (Left) and SN-type (Right) oocytes from PMSG-primed WT and Nat10-ZcKO mice in (**g**), and quantification of H3K9me3 intensity in (**h**). Scale bar, 10 μm. The box indicates the upper and lower quantiles, the thick line in the box indicates the median and whiskers indicate 2.5th and 97.5th percentiles. Data are presented as the mean ± SEM, $n = 3$; *$p < 0.05$ by two-tailed *Student's t-test*. NSN, $p = 0.0318$; SN, $p = 0.0270$. $n = 3$ biologically independent samples were included in each group (**a, c, e, g**). Source data are provided as a source data file.

meiotic maturation failed to be cleared in the Nat10-ZcKO MII oocytes (Fig. 7j).

To determine which transcripts were susceptible to degradation in the presence of Nat10, we defined a total of 2011 transcripts down-regulated in WT oocytes during the GV-MII transition and 1206 transcripts down-regulated in Nat10-ZcKO oocytes through GV-MII transition. Interestingly, a large fraction of transcripts (1416 among 2011 transcripts in total) did not overlap with down-regulated transcripts in Nat10-ZcKO oocytes, suggesting that they were not timely degraded but aberrantly accumulated in Nat10-ZcKO oocytes (Fig. 7k). The degradation trend profiling verified that the 1416 transcripts indeed displayed a decreased degradation speed in the absence of Nat10 (Fig. 7l). Of note, a total of 442 genes displayed abnormal alternative splicing patterns (Supplementary Fig. 4h, i). Taken together, these studies suggest that Nat10 is essential for the establishment of normal maternal transcriptome by the timely degradation of selective maternal mRNAs in the MII oocytes.

## Hairpin Adaptor-Poly(A) Tail length (HA-PAT) assay, a simple, cost-efficient, and sensitive method for validation of poly(A) mRNA degradation

CCR4-NOT deadenylase complex mediated poly(A) mRNA degradation has been well-documented to be responsible for maternal mRNA degradation during the MZT[37,44,45]. Indeed, we discovered that ~65% (782/1196) of transcripts up-regulated in Nat10-ZcKO MII oocytes were also accumulated in the Cnot6l (CCR4-NOT subunit) KO MII oocytes (Fig. 8a). Intriguingly, these up-regulated genes were especially enriched in ribosomal subunit components that were normally degraded through poly(A) tail-shortening mediated mRNA decay via the CCR4-NOT complex, a phenotype that was also observed in CCR4-NOT-deficient oocytes (Fig. 8b)[37,46]. Further examination by qPCR analysis revealed that the mRNA levels of Cnot7 and Btg4 were significantly decreased, whereas the Cnot6l mRNA levels reduced but without statistically significant difference between the WT and KO MII oocytes (Fig. 8c). Strikingly, further immunoblotting analyses demonstrated that the protein levels for CNOT6L, CNOT7 and BTG4 were vastly declined in MII oocytes from Nat10-ZcKO mice (Fig. 8d), suggesting that Nat10 has a profound impact on both the transcriptional and translational regulation. This evidence led us to postulate that Nat10-driven maintenance of CCR4-NOT deadenylase activity is required for the maternal poly(A) mRNA clearance and consequently the establishment of the oocyte maternal transcriptome.

We thus next tested whether the accumulated transcripts harbor long poly(A) tails in Nat10-ZcKO MII oocytes. Currently, two methods are commonly adopted for Poly(A) tail length (PAT) examination, including Ligase-mediated PAT (LM-PAT) and extension PAT (ePAT)[47–49]. The original LM-PAT method relies on oligo(dT)$_{12-18}$ hybridization and T4 ligation, followed by oligo-(dT) anchor primer-mediated PCR amplification[48,49]; ePAT exploits a hybridized oligo(dT) anchor primer as a DNA template for Klenow enzyme-mediated 3′ extension ensued by PCR amplification[47]. Given the limited availability of Nat10-null MII oocytes, we initially tried both methods using only 5–10 oocytes as

input materials; however, the PCR amplification either failed or produced very weak, unsatisfactory results, presumably owing to the low sensitivity of both methods when dealing with low-input samples.

Therefore, we next sought to develop a PAT assay that can circumvent the drawbacks of previous approaches requiring large-scale input materials (Fig. 8e). After several rounds of optimization, we termed this method the Hairpin Adaptor-Poly(A) Tail length (HA-PAT), which is more sensitive, low-cost and time-saving than previous methods[47,49]. HA-PAT utilizes an exquisitely designed hairpin adaptor that can self-hybridize plus an extended oligo (dT)$_8$ with two additional degenerate W nucleotides at the 3′ end (Fig. 8e). The loop sequence is connected through a C3 spacer, which provides a sufficiently flexible linker to enhance the self-hybridization of stem sequences. In practice, this strategy was optimized to facilitate specific anchoring at the 3′ end of poly(A) tail and to reduce the random dT hybridization background. Moreover, given that the 3′ ends of poly(A) mRNA tails dominate with two U nucleotides in the oocytes[7], the two degenerate "WW" nucleotides will further capture and stabilize the binding between the hairpin adaptor and the "U"-terminated poly(A) tails. Therefore, this single hairpin adaptor not only provides an anchor primer for reverse transcription but also harbors the full reverse primer sequence for subsequent cDNA PCR amplification.

Thanks to its integral design strategy, the total reagent cost and hands-on time were significantly reduced compared with the other two common methods (Supplementary Fig. 5a). To test its efficiency, we selected two known genes (*Rpl35a* and *Chchd2*) that accumulated in MII *Nat10-null* oocytes and the house-keeping *Gapdh* gene as a control. As shown in Supplementary Fig. 5b, using 5 oocytes as input material without PCR preamplification, neither LM-PAT nor ePAT could detect *Rpl35a* and *Chchd2* genes. In contrast, our HA-PAT assay yielded clear, differential smear bands between WT and Nat10-null oocytes, suggesting that the HA-PAT approach is more sensitive in detecting low-input mRNA samples (Supplementary Fig. 5b). Under PCR preamplification for 16 cycles using five oocytes, LM-PAT showed strong signals for the smear poly(A) tails, whereas ePAT failed to produce satisfactory results. Nonetheless, our HA-PAT method outperformed the other two methods since it gave rise to stronger and clearer bands (Supplementary Fig. 5b). This evidence validated the sensitivity and efficiency of the LM-PAT assay in detecting poly(A) tail length using a minute amount of input RNA material from single oocytes.

Taking advantage of the HA-PAT approach, we next selected a panel of representative genes showing up-regulation in each GO cluster identified by mini-bulk SMART-Seq2 for validation. We collected five hormone-primed oocytes at GV and MII stages for each sample and executed 16 cycles of PCR amplification. As shown in Fig. 8f, g, all the poly(A) tails of the seven genes were lengthened in MII *Nat10-null* oocytes, as judged by the elevated, smear PCR bands. Additionally, as cross-validation, we further performed LM-PAT, which corroborated the similar findings (Supplementary Fig. 6). Taken together, these results indicate that Nat10-mediated poly(A)-tail shortening and the resulting mRNA degradation sculpted the transcriptome of MII oocytes.

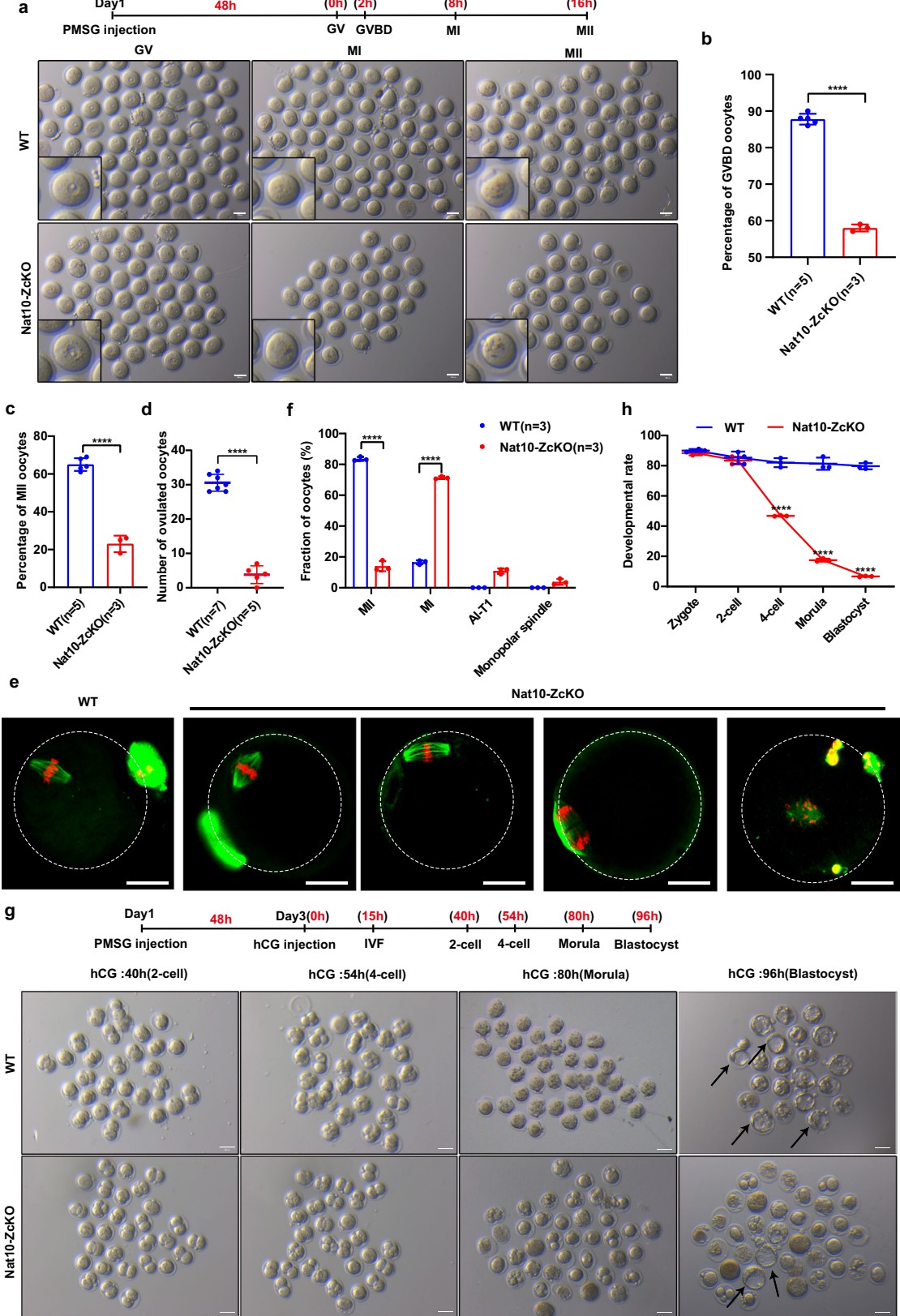

## Evidence that maternal Nat10 is translationally required for oocyte development

Nat10 has been shown to be associated with enhanced translation in HeLa cells in vitro[18]. We next examined if and how Nat10 impacted mRNA translation in a different cellular context. Given the limited availability of oocytes, we decided to generate stable, Cre-inducible

cell lines by exclusively utilizing our established Nat10[lox/lox]; Ubc-CreERT2 mice. We followed a "3T3" protocol and successfully generated two stable, 4-hydroxytamoxifen (4'-OHT)-inducible, mouse embryonic fibroblast (MEF) cell lines from E13 embryos (Nat10[lox/lox]; Ubc-CreERT2) after 2-month in vitro culture (Fig. 9a)[50]. Immunoblotting and qPCR analyses demonstrated that Nat10 protein and mRNAs

**Fig. 6 | Postnatal Nat10 ablation led to defective oocyte meiotic maturation.**
**a** The gross morphology of oocytes collected at the time points as indicated for GV (0 h), and cultured in vitro for MI (6 h) and MII (16 h) from PMSG-primed WT and Nat10-ZcKO females. Scale bar, 100 μm. **b, c** Percentage of oocytes at GVBD in **b** and MII in **c** after the release of GV oocytes cultured in IBMX-containing medium from PMSG-primed WT and Nat10-ZcKO females. Data are presented as mean ± SEM, *n* = 3. **\**p* < 0.01, ***\*\**p* < 0.001 by two-tailed *Student's t-test*. **d** Average numbers of superovulated oocytes at MII from WT (30.57 ± 0.92) and Nat10-ZcKO (3.8 ± 1.15) mice following PMSG and hCG injection in vivo. Data are presented as the mean ± SEM, *n* = 5. ***\*\**p* < 0.001 by two-tailed *Student's t-test*. **e** and **f** Immunofluorescence staining images of superovulated oocytes collected at 16 h after hCG injection by α-TUBULIN staining. Oocytes with MI arrest, Anaphase-to-telophase arrest in prophase I (AI-TI), and aberrant spindles were observed **e** and counted **f** in Nat10-ZcKO mice. Scale bar, 40 μm. Data are presented as the mean ± SEM, *n* = 3. ***\*\**p* < 0.001 by two-tailed *Student's t-test*. **g, h** Representative gross morphology of preimplantation embryos at various stages as indicated derived from superovulated WT and Nat10-ZcKO oocytes (after hCG priming) fertilized with WT sperm in (**g**). Arrows point to the blastocysts; a quantitative comparison of the average numbers of preimplantation embryos at varied stages was shown in (**h**). Scale bar, 100 μm. Data are presented as the mean ± SEM, *n* = 5. ***\*\*\**p* < 0.0001 by two-tailed *Student's t-test*. *n* = 3 biologically independent samples were included in each group (**a, e, g**). Source data are provided as a source data file.

were vastly declined upon OHT induction, respectively (Fig. 9b, c), indicative of the successful establishment of two stable cell lines. Apparently, Nat10 depletion significantly reduced cell proliferation without any effect on cell apoptosis, as evidenced by the Ki67 staining and the CCK8 assays (Fig. 9d–h). Reciprocally, rescue by Nat10 over-expression in Nat10-inducible KO MEF cells recovered and enhanced the cell division (Fig. 9f–h). Noteworthily, we consistently observed that longer induction of Nat10-inducible KO MEF cells by 4′-OHT treatment led to complete cell cycle arrest and resultant cell death, suggesting that Nat10 is essential for cell survival. The permissive cell death in Nat10-inducible KO MEF cells contrasted with the viability observed in *Nat10-null* HeLa cells[18]. Next, we pursued to explore how translation was impacted in Nat10-inducible KO MEF cells. We collected similar numbers of MEF cells without and with 4′-OHT treatment for three days for polysome profiling. Consistently, we found that Nat10-inducible KO cells displayed lower levels of mRNA-bound polysomes than non-treated MEF cells, indicating the translational efficiency was repressed in the Nat10-null MEFs (Fig. 9i).

Next, we optimized a low-input ribosome profiling in conjunction with ligation-free library preparation and sequencing (Ribo-seq) protocol using ~100 oocytes at GV and MII stages. Compared with the published Ribo-seq by Sha et al. using 500 oocytes[37], our Ribo-seq protocol detected comparable numbers of translated genes and global translational patterns (Supplementary Fig. 7a–c). A snapshot of the IGV browser tracks displayed characteristic RPF (ribosome-protected fragments) signatures at the exon regions for randomly selected genes by both approaches (Supplementary Fig. 7c). On average, our method detected ~9425 genes in GV and MII oocytes (Supplementary Fig. 7d). The mapped reads showed high coverage in the coding sequence (CDS) and low coverage in the 3′UTRs (Supplementary Fig. 7e). The 3-nt periodicity (Supplementary Fig. 7f) and good correlations were also observed in all samples (Supplementary Fig. 7g). Ribosome pro-filing unveiled that the RPFs for a total of 2766 transcripts were up-regulated whilst 2908 transcript RPFs down-regulated in Nat10-ZcKO MII oocytes (Fig. 10a), suggesting Nat10 is translationally required in the MII oocytes.

To evaluate how the global translatome was impacted at the transcript level during GV-MII transition, we further overlapped the RPF translatome and the transcriptome of MII oocytes, and strikingly, we observed that 491 gene transcripts significantly accumulated in Nat10-ZcKO MII oocytes displayed up-regulated RPF enrichment in translatome (Fig. 10b, c). GO analysis revealed that those gene tran-scripts were predominantly enriched in translation, RNA processing and blastocyst formation (Fig. 10d–f). Noteworthily, the majority of the genes displaying the up-regulated levels in both translatome and transcriptome in Nat10-ZcKO MII oocytes exhibited increased TE (Translational Efficiency = RPF/mRNA) (Fig. 10g–i). Hence, the increased expression levels for those aberrantly accumulated maternal transcripts in the Nat10-ZcKO MII oocytes presumably caused the defective GV-MII transition (Fig. 10b). However, it is worth noting that the overall TE levels decreased in Nat10-ZcKO MII oocytes, suggesting Nat10 loss caused global translational repression in MII oocytes

(Fig. 10j, k). Altogether, these results signify that Nat10 is a pivotal factor essential for translational regulation in the meiotic oocytes.

### ac4C modification enhances the translation efficiency for a cohort of transcripts enriched in translation and chromosome segregation in mouse oocytes

To explore whether there exists a causative relationship between Nat10-mediated ac4C modification and translational regulation, we conducted a low-input ac4C RNA immunoprecipitation sequencing (acRIP-seq) using pools of ~2000 WT GV oocytes following optimiza-tion of a published protocol[38,51]. This revealed that a total of 6188 transcripts likely harbor the ac4C modification (ac4C/IgG>2) in the oocytes, although this approach cannot pinpoint the accurate posi-tions of cytosine acetylation due to the technical limitation of ~200 bp mRNA fragment resolution. GO enrichment revealed that the tran-scripts with ac4C modification (ac4C+) were mostly enriched in cell cycle and mRNA processing pathways (Supplementary Fig. 8a). We next verified the validity of our acRIP-seq data by RIP-PCR for genes that have been shown to harbor ac4C modification in HeLa cells published[18]. We selected Bst2, Fus, and Nomo as the positive control (ac4C+), and Eef1a1 and Gapdh as the negative control (ac4C−). Our method clearly detected the bands for ac4C+ gene transcripts with ac4C antibody but not IgG isotype control. For ac4C− transcripts as negative controls, there is no band seen in either ac4C or IgG beads. Moreover, the band intensity in the ac4C supernatant for ac4C+ genes was also present but with reduced levels, suggesting the pulldown efficiency by the ac4C antibody was not 100% efficient (Supplementary Fig. 8b). This result confirmed the validity and sensitivity of our acRIP data. Next, we performed RIP-PCR to verify whether Cnot6l, Cnot7, and Btg4 harbor ac4C modification under the same condition. This assay revealed that all three gene transcripts were clearly modified by ac4C, albeit with reduced signal seen for Cnot6l as compared with Cnot7 and Btg4 (Supplementary Fig. 8c, d).

Next, we jointly analyzed the overlapping gene transcripts among Smart-seq2, Ribo-seq, and acRIP-seq data, which revealed that, among a total of 2647 up-regulated transcripts in *Nat10-null* MII oocytes, 185 (41.11%) ac4C+ gene transcripts and 265 (58.89%) ac4C− transcripts showed dysregulated RPFs enrichment (Supplementary Fig. 9a). GO analyses of ac4C+ transcripts implied that they were mainly related to translation, RNA splicing and chromosome separation (Supplementary Fig. 9b). Of note, it appeared both the mRNA (Supplementary Fig. 9c–e) and the RPFs enrichment levels (Supplementary Fig. 9f–h) were higher for the ac4C− gene transcripts, but not for the ac4C+ transcripts, when compared between the WT and Nat10-null MII oocytes. We didn't observe any difference in terms of the TE between ac4C+ and ac4C− gene transcripts (Supplementary Fig. 9i–k). Further hierarchical clustering for both ac4C+ and ac4C− genes clearly showed a cohort of aberrantly accumulated transcripts with enhanced RPFs enrichment and improved TE for both ac4C+ and ac4C− transcripts (Supplementary Fig. 9l). Together, these results suggest that while Nat10-deposited ac4C modification does enhance a cohort of mRNA translation, it likely functions not alone, but synergistically with other

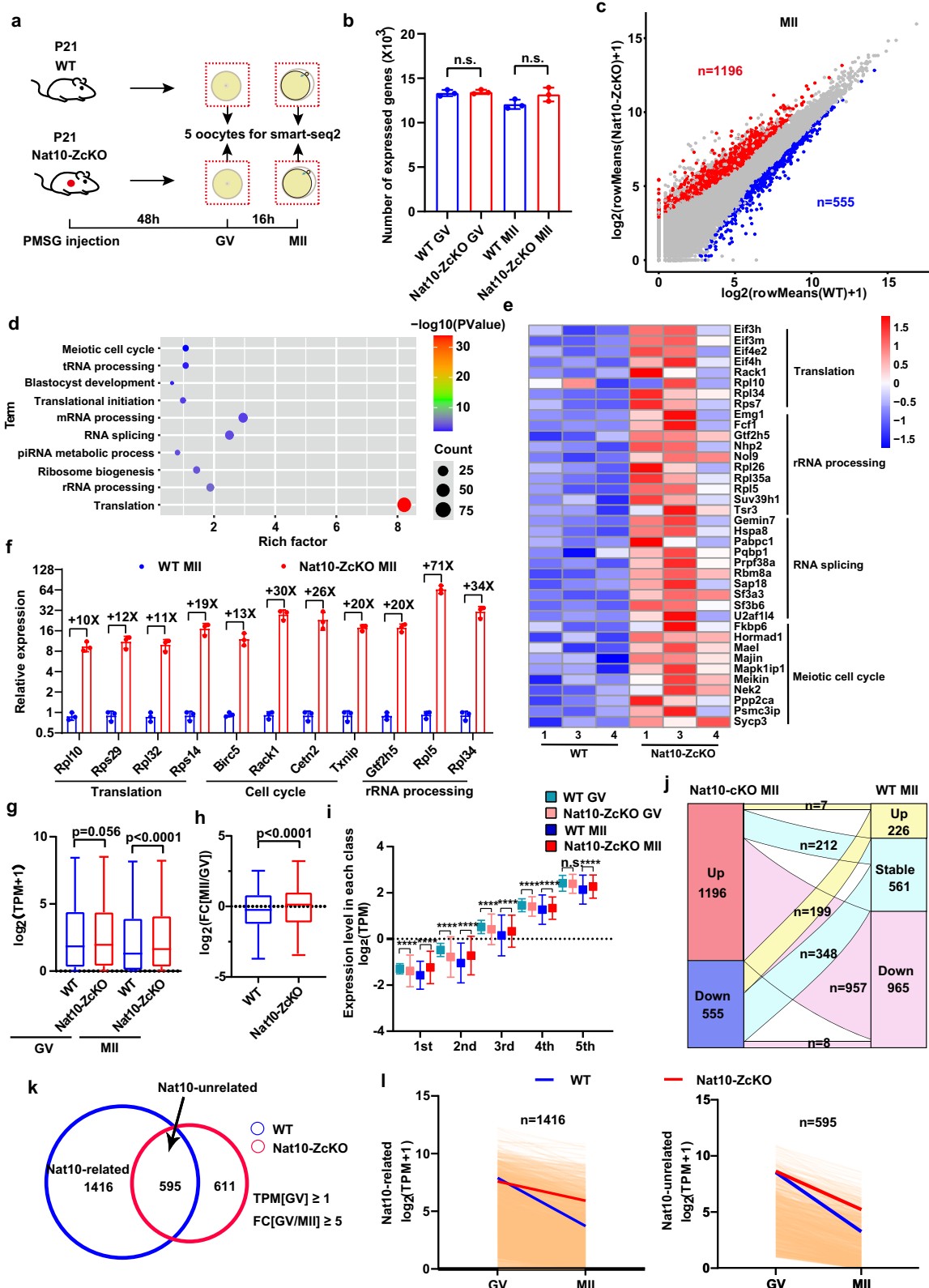

unidentified Nat10-dependent factors, to jointly modulate the target mRNA translation in the mouse oocytes.

## Discussion

Dysregulated expression of Nat10 has been recently linked to numerous diseases, such as Hutchinson–Gilford progeria syndrome (HGPS)[52], epithelial ovarian cancer[53], breast cancer[19], and spermatogenesis[22], and

is most recently implicated in oocyte in vitro maturation (IVM)[54]. Mature oocytes are specialized, transcriptionally inert cells that almost exclusively contribute to the cytoplasm of zygotes when fertilized by sperm. Therefore, all the early developmental events of preimplantation embryos occur at the post-transcriptional level and are dependent on the stored RNA content derived from oocytes, called the maternal transcriptome[6,36,39,40]. In mammals, a hallmark of gene expression for

**Fig. 7 | Mini-bulk SMART-seq2 analyses identified the dysregulated maternal transcriptome in Nat10-ZcKO oocytes. a** A diagram showing mouse oocyte sample collection for mini-bulk SMART-seq2. **b** Bar graph showing the numbers of transcripts detected in WT and Nat10-ZcKO oocytes at GV and MII stages (TPM ≥ 1). The box indicates the upper and lower quantiles, the thick line in the box indicates the median, and the whiskers indicate the 2.5th and 97.5th percentiles. Data are presented as mean ± SEM, *n* = 3 biologically independent samples. n.s., non-significant by two-tailed *Student's t-test.* **c** Scatter plot of mini-bulk SMART-seq2 data showing differentially expressed genes (DEGs) in Nat10-ZcKO MII oocytes. Red color: Up-regulated; Blue color: Down-regulated; Cutoff: fold change (FC) ≥ 2, adjusted *p* < 0.05. **d** Gene Ontology (GO) enrichment analysis of up-regulated genes in Nat10-ZcKO MII oocytes (Cutoff: FC ≥ 2, adjusted *p* < 0.05). **e** Heatmap of representative genes from four major functional GO categories showing up-regulated expression in Nat10-ScKO MII oocytes. **f** Bar plots showing the qPCR analyses of relative mRNA expression levels for a panel of up-regulated genes identified by mini-bulk SMART-seq2. Data are presented as the mean ± SEM, *n* = 3 biologically independent samples, and analyzed by two-tailed *Student's t-test.* **g** and **h** Box plots showing the relative expression levels of the transcripts at the GV and MII stages as indicated in (**g**). The box indicates the upper and lower quantiles, the thick line in the box indicates the median, and the whiskers indicate the 2.5th and 97.5th percentiles. Data are presented as the mean ± SEM, *n* = 3. ****p* < 0.0001 by two-tailed *Student's t-test.*; and the relative fold changes of mRNA levels in (**h**). The box indicates the upper and lower quantiles, the thick line in the box indicates the median and the whiskers indicate 2.5th and 97.5th percentiles. Data are presented as the mean ± SEM, *n* = 3. ****p* < 0.0001 by two-tailed *Student's t-test.* **i** Box plot showing gene expression levels at the GV and MII stages with genes divided into 5 bins according to their relative expression abundance in the WT MII oocytes. Data are presented as the mean ± SEM, *n* = 3 biologically independent samples. ****p* < 0.0001, n.s., non-significant by two-tailed *Student's t-test.* **j** Sankey diagram showing the overlapping of the DEGs (1196 up-regulated vs. 555 down-regulated) with genes exhibiting up- [FC(MII/GV) ≥ 2, *p* < 0.05], down- [FC(MII/GV) ≤ −2, *p* < 0.05], or stable expression patterns in WT MII relative to GV stage oocytes (TPM ≥ 1). **k** Venn diagram showing the overlapping of down-regulated transcripts between WT MII oocytes relative to GV oocytes (2011, Cutoff: TPM ≥ 1, FC[GV/MII] ≥ 5), and Nat10-ZcKO MII oocytes relative to GV oocytes (1206, Cutoff: TPM ≥ 1, FC[GV/MII] ≥ 5). The total 595 overlapping transcripts represent those that were concurrently down-regulated in both WT MII and Nat10-ZcKO MII oocytes. In other words, they were degraded regardless of Nat10 presence (Nat10-unrelated). **l** Degradation trend patterns of maternal transcripts during the GV-MII transition in WT and Nat10-ZcKO oocytes. Each light-yellow line represents the expression levels of one gene, and the blue and red lines represent the median expression levels in WT and Nat10-ZcKO, respectively. Source data are provided as a source data file.

the maternal transcriptome is characterized by the uncoupling of transcription and translation—some accumulated mRNA species are immediately translated to support oocyte growth, while a large stock of other species is stabilized as ribonucleoprotein particles (RNPs) in a translationally inactive state, in growing follicles[6,43,44]. When fully grown GV oocytes resume meiosis in a process called meiotic maturation, however, the situation substantially changes—many previously active mRNAs become translationally silent, whereas some previously "dormant" mRNA species are translationally reactivated. A massive wave of RNA elimination is considered as a hallmark and a major driving force underlying oocyte-to-embryo transition (OET). It has been estimated that up to ~40% poly(A) RNA is degraded during the GV-to-MII oocyte transition; however, the molecular mechanisms underlying this RNA remodeling event are not well understood[6,43,44].

Post-transcriptional RNA modifications constitute an exciting layer of so-called "epi-transcriptome" that modulates gene expression without altering RNA sequences. Recent evidence showing N6-methyladenosine (m6A) modification abundantly present in mRNA species is an attractive model that selectively marks any mRNA for degradation at the post-transcriptional level. For instance, m6A is specifically deposited by a large, heterogeneous multiprotein complex, known as the m6A "writer", which comprises a core catalytic member of METTL3, along with METTL14, WTAP and KIAA1429 (VIRMA) cofactors[55]. Genetic evidence in conjunction with m6A antibody-mediated RNA-immunoprecipitation sequencing (m6A-seq, or MeRIP-seq) showed that the m6A mark specifically decorates the 3'UTR close to the STOP codon region, and defective MZT transition is tightly associated with reduced m6A levels in m6A-deficient oocytes, suggestive of an essential role of m6A in active RNA degradation through the MZT[5,17,38,56].

In comparison to the high prevalence of m6A modification (~0.5%) in mRNAs, the presence and function of ac4C modification in mammalian mRNAs are currently controversial[57]. An initial study identified that ac4C tends to enrich in the 5'UTR region in thousands of coding mRNAs, through ac4C antibody-mediated immunoprecipitation and sequencing (acRIP-seq) in HeLa cells[18,58]. However, surprisingly, a later study unveiled not even a single ac4C site in mammalian mRNAs through a convincing chemistry-catalyzed strategy with quantitative and nucleotide resolution[59,60]. Importantly, as a positive control, they successfully identified the previously known ac4C sites in both rRNA and tRNAs, as well as low levels of a few hundred ac4C acetylation sites in Nat10-overexpressing human cells, implying that their adopted

method is sensitive and feasible for detecting ac4C modification[59,60]. Hence, the ac4C modification levels are, if present, extremely low in human mRNAs. Through acRIP-sequencing, we have found that ac4C modification is abundantly present in the maternal oocyte transcripts, which are functionally involved in translation and RNA processing (Supplementary Fig. 9). While the global mRNA translation was repressed in Nat10-null oocytes, the aberrantly accumulated ac4C+ transcripts, including many ribosome-processing subunit transcripts, exhibited elevated protein translation, which was clearly verified by our *in-house* optimized HA-PAT and Ribo-seq approaches (Figs. 7, 8, 10). In addition, we discovered that the key components of CCR4–NOT complex, including Cnot6l, Cnot7, and Btg4, all harbor the ac4C modification, and were subjected to transcriptional and translational repression in the Nat10-null oocytes, causing the delayed maternal transcript degradation owing to reduced activity of CCR4–NOT complex and the resultant dys-regulated maternal transcriptome. These data suggest that NAT10 is functional through oocyte maturation, at least in part, through the deposition of ac4C modification at both the transcriptional and translational levels.

NAT10 is a protein highly conserved from *E. coli* and yeast to mammalian species and is the solely known enzyme responsible for ac4C modification on RNA substrates. It is a relatively large protein that comprises four distinct domains, including DUF1726, Helicase, GNAT and tRNA binding domain[25]. In this study, we employed two Cre deletor mouse models to cross with floxed *Nat10* alleles to attain germline-specific or inducible *Nat10* KO mice, yielding full *Nat10-null* mouse offspring without any of the four domains owing to premature frame-shift reading. Consistent with the high expression levels of Nat10 in the female germline, pre-meiotic ablation of Nat10 caused apparent meiotic arrest at the pachytene stage, resulting in premature ovarian failure and female infertility in adults (Fig. 2). We showed that this defect was likely caused by the deficient DSB repair as judged by the high persistent γH2AX intensity and enhanced RPA2 remnant (Fig. 3). This conclusion agrees with a previous study showing that DNA-damaging agents induced Nat10 expression in a dosage- and time-dependent fashion[61]. Nat10 depletion in growing oocytes of primary follicles disrupted the NSN-SN transition of GV oocytes and damaged meiotic maturation, as evidenced by the aberrant GV to MII transition[11]. By optimized mini-bulk SMART-seq2 analyses, we discovered that a large number of genes enriched in cell cycles and DNA transcription were dysregulated in Nat10-null GV oocytes (Supplementary Fig. 3). In particular, we revealed that many genes destined for degradation in the MII oocytes were aberrantly accumulated in *Nat10-*

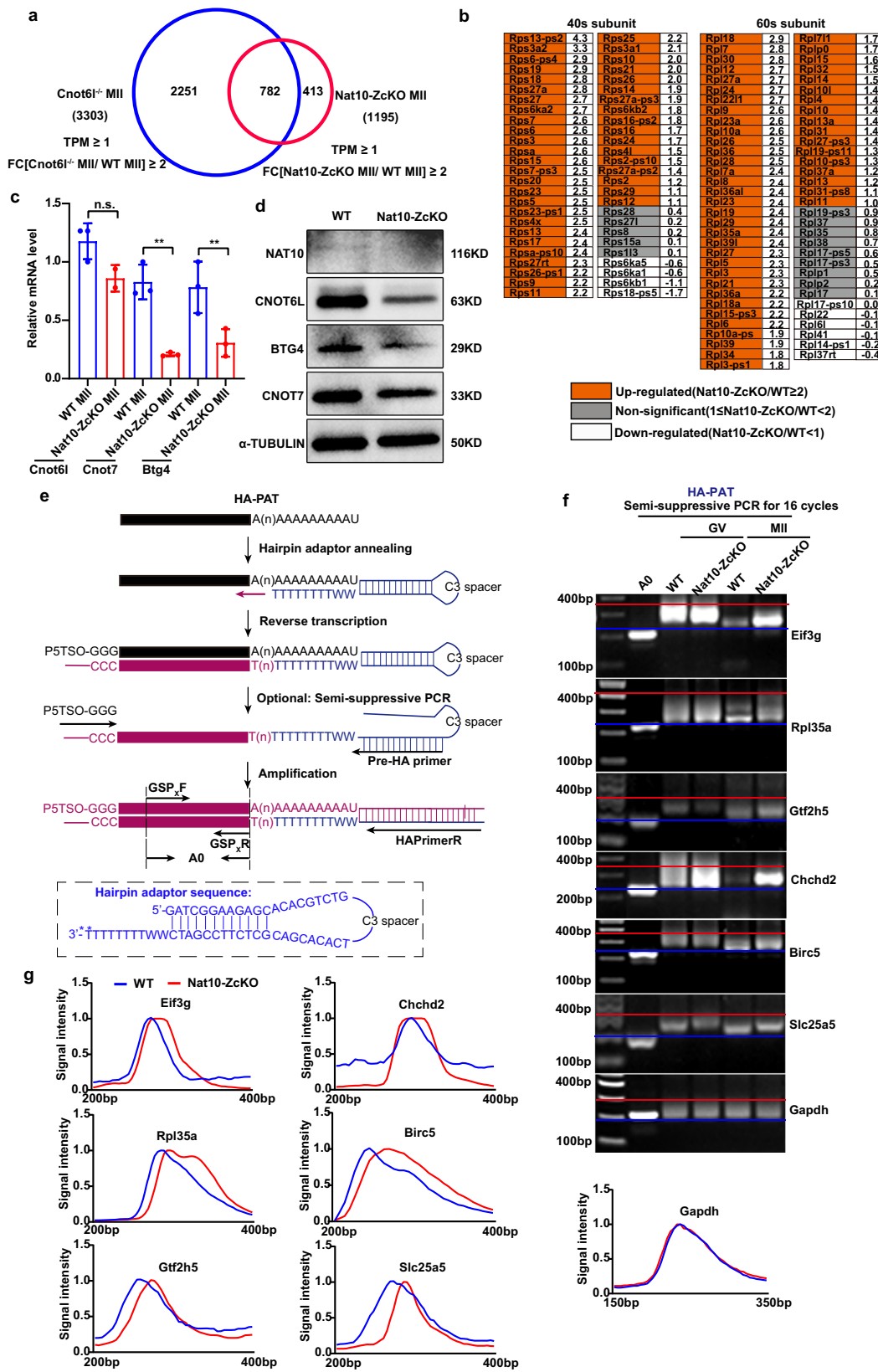

_null_ MII oocytes, which was corroborated by an in-house developed HA-PAT approach.

Further evidence showed that Nat10 deletion caused the down-regulated expression of important members of the CCR4-NOT complex, at least in part, at the transcriptional level[37,44]. These studies demonstrate that NAT10 is transcriptionally essential to maintain transcriptomic homeostasis to facilitate oocyte growth and maturation. It is worth to note that there is a delicately designed, low-put poly(A) tail length sequencing approach recently published using human oocytes, known as PAIso-seq[62,63]. It can accurately capture not only the poly(A) length but also the non-A nucleotide variants in the poly(A) region by utilizing full-length cDNA sequencing with Pacbio.

**Fig. 8 | Hairpin Adaptor-Poly(A) Tail length (HA-PAT) assay validated the deficient maternal mRNA decay in Nat10-ZcKO MII oocytes. a** Venn diagram showing the overlapping of transcripts that were stabilized during GV-to-MII transition in Cnot6l−/− and Nat10-ZcKO MII oocytes (FC = [WT MII/Nat10-ZcKO MII]≥2, $p$ < 0.05). **b** Fold change of relative expression levels of transcripts encoding ribosomal protein subunits in Nat10-ZcKO relative to WT oocytes at the MII stage. The values of log2(FC[Nat10-ZcKO/WT]) are listed in the right column. **c** qPCR results showing the relative levels of indicated transcripts (Cnot6l, Cnot7 and Btg4) in WT and Nat10-ZcKO oocytes at MII stage. Data are presented as the mean ± SEM, $n$ = 3. n.s., non-significant, **$p$ < 0.01 by two-tailed *Student's t-test*. Cnot6l, $p$ = 0.0906; Cnot7 $p$ = 0.0020; Btg4 $p$ = 0.0031. **d** Western blot displaying the NAT10, CNOT6L, CNOT7 and BTG4 protein levels in MII oocytes of WT and Nat10-ZcKO mice. α-TUBULIN was used as a loading control. $n$ = 3 biologically independent samples were included in each group. **e** A schematic illustration depicting the design strategy and the key steps for Hairpin Adaptor-Poly(A) Tail length (HA-PAT)

assay. The 1st strand of cDNA was synthesized with the hairpin adaptor (HA) primer in conjunction with a P5TSO primer containing three "G", via a mechanism of "template-switching". GSP, Gene-specific primer; A0, the PCR product resulting from the amplification with a gene-specific pair of GSPxF and GSPxR primers; GSPxR primer was designed against an mRNA's 3′ terminals preceding the poly(A) sequence; poly(A)-containing PCR products were amplified with GSPxF and fixed HAPrimerR primers. The full sequence for hairpin adaptor (HA) is listed at the bottom. W indicates degenerate nucleotides (A or T); * The asterisk indicates the phosphorothioate modification. **f, g** HA-PAT assay results showing changes in poly(A)-tail lengths of indicated transcripts in WT and Nat10-ZcKO oocytes at GV and MII stages. Experiments were performed in triplicates; a representative image is shown in the 2% agarose gel in **f** and the length distribution is shown in the densitometric curves in (**g**). $n$ = 3 biologically independent samples were included in each group (**d, f**). Source data are provided as a source data file.

However, it is considerably expensive and relatively not affordable by individual labs. Finally, we provided multi-omics data analyses using Smart-seq2, acRIP-seq and Ribo-seq datasets, and revealed that Nat10 is both transcriptionally and translationally required for the activity of CCR4-NOT complex, at least in part, by depositing the ac4C modification on the key transcript components (Cnot6l, Cnot7 and Btg4). Nonetheless, we should point out that while ac4C does improve the translation efficiency for a cohort of gene transcripts crucial for oocyte meiotic division, there are Nat10-dependent, but unidentified factors that function synergistically with ac4C to fine-tune the transcription and translation in the oocytes, which would be the goal of future studies for in-depth deciphering the mechanism of Nat10 functions in the germline.

In conclusion, we provide genetic and mechanistic studies showing that Nat10 is transcriptionally and translationally compulsory for the prophase I progression and meiotic resumption during oocyte growth and meiotic maturation. However, further experiments are urgently needed to elucidate the functional diversity of NAT10 and the causative relationship between the phenotypic outcome and the distinct domains of NAT10.

## Methods

### Mice
The floxed Nat10 (Nat10^lox/lox) alleles were generated by GemPharmatech Co., Ltd. Conditional *Nat10* knockout mice were achieved by crossing Nat10^lox/lox mice with *Stra8-GFPCre* and with *Zp3-Cre* mice to attain the Nat10-ScKO and Nat10-ZcKO offspring, respectively. *Nat10* KO MEF cells were achieved from the embryo of Nat10-Ubc mice by crossing Nat10^lox/lox mice with Ubc-CreERT2 KI mice. The Stra8-GFPCre knock-in (KI) mouse line was generated in Ming-Han Tong's Lab at the Shanghai Institute of Biochemistry and Cell Biology, Chinese Academy of Sciences. *Zp3-Cre* and Ubc-CreERT2 KI mice were obtained from Jackson Laboratory. All mice were from the C57BL/6J background and were bred in a specific pathogen-free (SPF) facility with a 12 h light/dark cycle and with free access to food and water. All animal experiments were approved by the Animal Care and Research Committee of the University of Science and Technology of China (USTC).

### Inducible *Nat10* KO cell lines and culture
We generated two tamoxifen-inducible, stable cell lines following a standard 3T3 protocol[50]. In brief, the pregnant female mice were euthanized, and the embryos at embryonic day 13.5 (E13.5) were carefully dissected. After removing somatic organs, the embryonic body was chopped into small pieces and digested with DMEM medium containing 0.25% trypsin-EDTA in a 37 °C water bath for 30 min. After filtering through a 70 μm cell strainer, the single cells were cultured in MEF medium (DMEM medium supplemented with 10% FBS (VivaCell, C04001) and 1% Penicillin-Streptomycin (Biosharp, BL505A)). The cells were subcultured and the medium was replenished every 3 days, with

the passage number recorded. Cells were grown in a 5% $CO_2$ cell culture incubator (Heal Force) at 37 °C. Two stable MEF lines were achieved after passage 23 following recovery of the MEF cells from the crisis around passage -10–14 (total time is ~2 months).

### Lentiviral transduction
A mouse Nat10 cDNA plasmid clone (EX-Mm12636-M45) was purchased from GeneCopoeia, Inc. and cloned into pCDH-CMV vectors in-frame with a FLAG tag. The lentiviral vector was co-transduced into 293T cells alongside a packaging vector psPAX2 and a helper vector pCMV-VSVG using LIP2000 transfection reagent (Biosharp, BL623B) for lentivirus production. Viral particles were collected to infect MEF cells with 5 μg/ml Polybrene (Solarbio, H8761). The infected cells were positively selected with puromycin (12.5 μg/ml) (Solarbio, P8230) for 48 h.

### Oocyte collection and in vitro culture
The 21-day-old female (P21) mice were injected with 5 IU of pregnant mare's serum gonadotropin (PMSG) (Ningbo Sansheng Pharmaceutical). After 48 h, the mice were euthanized and the oocytes at the GV stage were harvested in M2 medium (Nanjing Aibei Biotechnology Co., Ltd, M1250) and further cultured in M16 medium (Sigma, M7292) covered with mineral oil (Sigma, M5310) at 37 °C in 5% $CO_2$. Samples were imaged with a microscope (SZX7, Olympus).

### Superovulation and in vitro fertilization
For superovulation, 21-day-old female mice were injected with 5 IU of PMSG. After 46 h, the mice were injected with 5 IU of human chorionic gonadotropin (hCG) (Ningbo Sansheng Pharmaceutical). At post-hCG 16 h, the cumulus–oocyte complex (COC) was retrieved from the oviducts, and the oocytes were counted after digestion with 0.3% hyaluronidase (Sigma, H4272). For in vitro *fertilization* (IVF), superovulated female mice were euthanized 15 h after hCG injection, and the ampulla parts of the oviducts were collected in an HTF medium covered by mineral oil. COCs were released and fertilized with WT sperm in HTF for 30 min at 37 °C in a 5% $CO_2$ incubator and further cultured in KSOM medium in vitro (Sigma, MR-101).

### RNA isolation and real-time RT-PCR
Total RNA was isolated from mouse tissues with TRIzol Reagent following the manufacturer's instructions as described previously[64]. In brief, freshly collected or frozen tissues were homogenized in TRIzol reagent. For MEF cells, total RNA was isolated with a SPARKeasy Cell RNA kit (Shandong Sparkjade Biotechnology Co., Ltd., AC0205). The quantity and quality of RNA samples were determined by measurement using a NanoPhotometer N50 (Implen, Germany). The RNA samples with OD values of 260/280 ≥ 1.9 were selected for downstream analyses. Equal amounts of total RNA were loaded to synthesize cDNAs using the Hiscript III Reverse Transcriptase (Vazyme, R302-1).

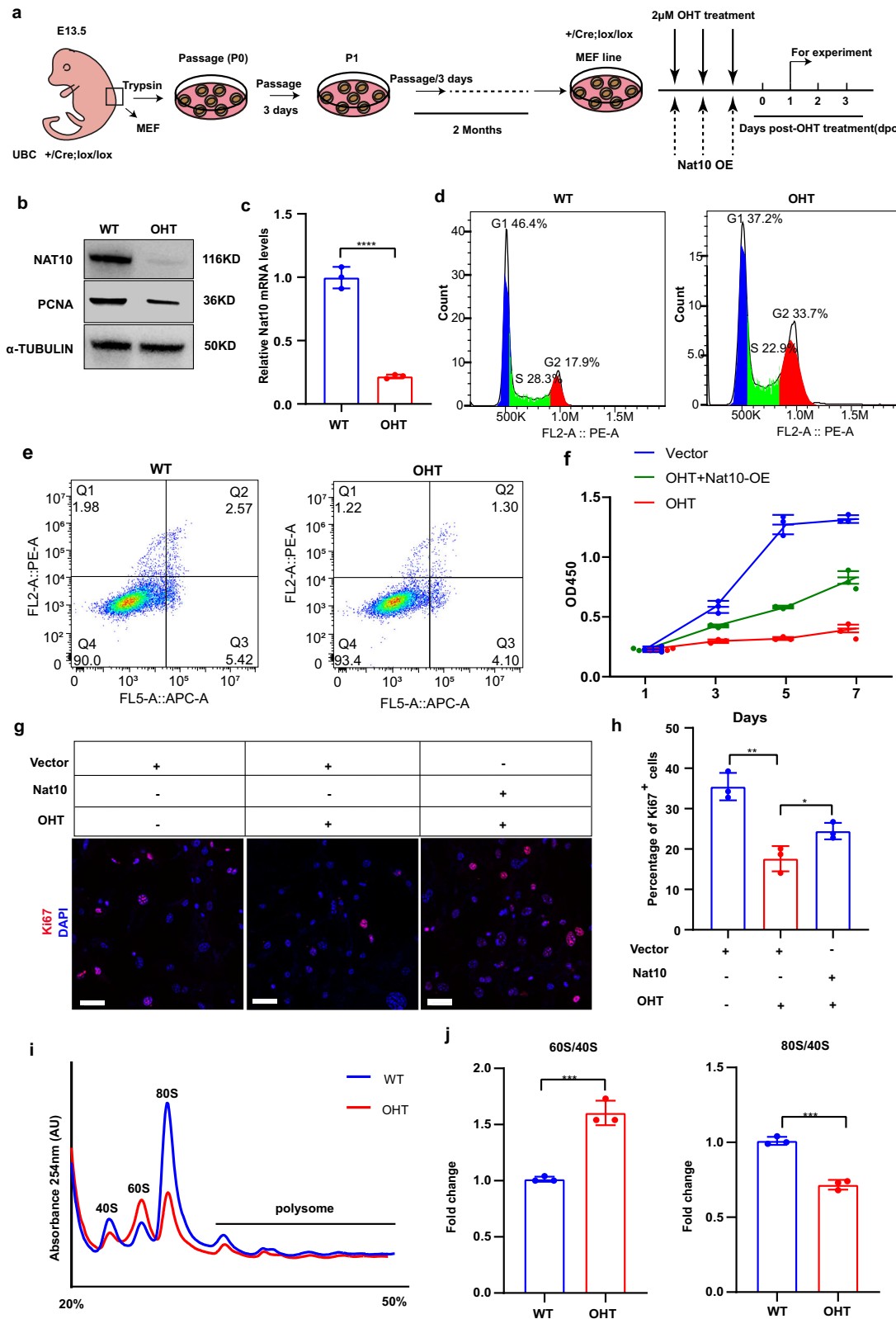

Quantitative PCR (qPCR) was performed using Hieff® qPCR SYBR Green Master Mix (Yeasen, 11201ES03) on a Q2000B Real-Time PCR machine (LongGene). For oocytes, 5–10 oocytes were lysed in 2 µl lysis buffer (0.2% Triton X-100 and 2 IU/µl RNase inhibitor) followed by reverse transcription and PCR pre-amplification for 8–16 cycles. The PCR products were diluted and used for qPCR. The primers are listed in Supplementary Data 1.

## Histological analysis

Hematoxylin & Eosin (HE) staining was performed following the standard procedure as described previously[64]. In brief, ovary samples were freshly collected, and fixed in Bouin's solution at room temperature overnight. Paraffin-embedded samples were cut into slides with 5 µm thickness. Slides were de-paraffinized with xylene and re-hydrated, followed by staining with HE. The slides were then dehydrated and

**Fig. 9 | Nat10 is translationally required for cell cycle progression. a** Schematic illustration showing the procedures for the generation of two stable, TMX-inducible MEF cell lines (Nat10[+/Cre, lox/lox]) from the Ubc-cre; Nat10[lox/lox] embryos following a 3T3 protocol (see the "Methods" section). Inducible Nat10 KO was achieved by 2 μM OHT treatment for three consecutive days. Rescue of Nat10 was attained by overexpression (OE) of WT Nat10 ORF plasmid. **b** Western blot displaying the NAT10 and PCNA protein levels in WT and OHT (Ubc-cre; Nat10[lox/lox]) MEF line. α-TUBULIN was used as a loading control. **c** Quantitative RT–PCR results showing the relative expression levels of Nat10 in WT and OHT MEF line. Data are presented as mean ± SEM, $n = 3$. ****$p < 0.0001$ by two-tailed *Student's t-test*. **d** Cell cycle analysis by flow cytometry between WT and OHT MEF cells. G1, $p = 0.0083$; S, $p = 0.0017$; G2/M, $p < 0.0001$; $p$-value are analyzed by two-tailed *Student's t-test*. **e** Annexin V apoptosis detection of WT and OHT MEF cells. Q4, Viable cells, $p = 0.2983$; Q3, Early apoptosis, $p = 0.2147$; Q2, Late apoptosis, $p = 0.3122$; Q1, Necrotic cells, $p = 0.7863$. $p$-value are analyzed by two-tailed *Student's t-test*. **f** CCK8

assay showing the cell proliferation rates among WT, OHT treatment and OHT plus Nat10 overexpression (OE) groups. Data are presented as the mean ± SEM, $n = 3$ biologically independent samples and analyzed by two-tailed *Student's t-test*. **g, h** Comparison of cell proliferation as visualized by Ki67 labeling in **g** and quantification in **h** among WT (empty vector), OHT treatment and Nat10 overexpression groups. Scale bar, 50 μm. Data are presented as mean ± SEM, $n = 3$. *$p < 0.05$, **$p < 0.01$ by two-tailed *Student's t-test*. Vector-OHT, $p = 0.0026$; OHT-Nat10, $p = 0.0340$. **i, j** The polysome profiling displaying the translational efficiency and ribosome assembly in MEF cells analyzed by sucrose density gradient centrifugation. The graphic curves showed the polysome profiles of MEF cells treated by mock (Blue) or OHT (Red) in **i**. Comparison of the ratios of 60S to 40S (Left) and of 80S to 40S (Right) in MEF cells with mock treatment (Blue) or OHT (Red) in (**j**). Data are presented as mean ± SEM, $n = 3$. ***$p < 0.001$ by two-tailed *Student's t-test*. 60S/40S, $p = 0.0008$; 80S/40S, $p = 0.0003$. $n = 3$ biologically independent samples were included in each group (**b, d, e, g, i**). Source data are provided as a source data file.

mounted with neutral resins. Images were taken on light microscopy (MShot) with an MSX2 camera.

## Western blot analysis
Samples were freshly collected and lysed in RIPA solution [100 mM Tris–HCl (PH7.4), 1% Triton X-100, 1% sodium deoxycholate, 0.1% SDS, 0.15 M NaCl, supplemented with Protease inhibitor cocktail]. Protein concentrations were determined using a BCA protein assay kit. All protein samples were run in 8% of denatured sodium dodecyl sulfate–polyacrylamide (SDS–PAGE) gel with Trelief® Prestained Protein Ladder (TSINGKE, TSP021), followed by wet-transfer to PVDF membranes. Subsequently, the membranes were blocked in 1XPBS with 5% non-fat milk and incubated with primary antibody followed by secondary antibody. Signals were visualized using an imaging system (SHST, Hangzhou, China). The following antibodies were used: rabbit anti-NAT10 (ZENBIO, 389412, 1:1000), mouse anti-PCNA (Proteintech, 60097-1-Ig, 1:2000), mouse anti-GAPDH (Proteintech, 60004-1-Ig, 1:10,000), rabbit anti-Tubulin (Proteintech, 11224-1-AP, 1:5000). rabbit anti-CNOT6/6L (Abcam, ab86209, 1:1000), rabbit anti-CNOT7 (ZEN-BIO, 382509, 1:1000), rabbit anti-BTG4 (Abcam, ab235085, 1:1000).

## Chromosome spreads analysis and immunofluorescent staining
For oocyte chromosome spreading analyses, newborn pups were sacrificed, and ovaries were dissected in hypotonic buffer [30 mM Tris–HCl (pH = 8.2), 50 mM sucrose, 17 mM trisodium citrate dihydrate, 5 mM EDTA (pH = 8.0), 1 mM dithiothreitol (DTT), and 1 mM phenylmethylsulfonyl fluoride (PMSF)] for 25 min. Next, ovaries were transferred to 100 mM sucrose, and single cells were released into sucrose solution using syringe needles. Single cells were spread and fixed in 1% PFA solution containing 0.15% Triton X-100 on slides, followed by washing with 0.4% Photo-Flo. For immunofluorescent staining, cells were permeabilized with 0.3% Triton X-100, blocked with 5% normal goat serum (Solarbio, SL038) in PBST, and incubated with primary antibodies diluted in blocking solution at 4 °C overnight. Antibodies used were as follows: mouse anti-SYCP3 (Abcam, ab97672, 1:1000), rabbit anti-SYCP1 (Abcam, ab15090, 1:1000), rabbit anti-SYCP3 (Proteintech, 23024-1-AP, 1:200), mouse anti-γH₂AX (Millipore, 05-636, 1:1000), rabbit anti-RPA2 (Proteintech, 10412-1-AP, 1:400). After washing in 1XPBS (0.1% Tween) for three times, the samples were incubated with TRITC-conjugated Goat Anti-Rabbit IgG (Proteintech, SA00007-2) or 488-conjugated secondary antibodies (Proteintech, gb2AF488).

For ovarian immunofluorescent staining, ovaries were fixed in 4% paraformaldehyde (PFA) overnight at 4 °C on a rocker. Slides were dehydrated with 10% and 20% sucrose for 2 h each. Ovary samples were cut into 8 μm slides. For immunofluorescence staining of MEF cells and oocytes, they were fixed in 4% PFA for 30 min. The primary antibodies used were as follows: rabbit anti-NAT10 (Proteintech, 13365-

1-AP, 1:400), rabbit anti-NAT10 (ZENBIO, 389412, 1:500), rabbit anti-Nucleophosmin (Abcam, ab10530, 1:1000), rabbit anti-H3K4me3 (Abclonal, A2357, 1:200), rabbit anti-α-Tubulin (Proteintech, 11224-1-AP, 1:200), and rabbit anti-KI67 (Servicebio, GB111141, 1:400). Slides were imaged by a Leica THUNDER Imager Live Cell with a K5 camera driven by the Leica Application Suite Software. Image processing was performed by ImageJ software.

## Poly(A)-tail (PAT) length assay
For the *Hairpin-Adaptor PAT* (HA-PAT) assay, 10 denuded oocytes were freshly lysed in 2 μl lysis buffer (0.2% Triton X-100 and 2 IU/μl RNase inhibitor). The hairpin adaptor was pre-annealed in 1Xoligo annealing buffer (50 mM Tris, pH 8.0, 50 mM NaCl, 1 mM EDTA). Poly(A)-tail mRNAs in the oocyte lysates were denatured at 72 °C for 3 min and then hybridized with the annealed hairpin adaptor at 25 °C for 10 min. The reverse transcription mix contained 100 U of SuperScript IV, 10 U of RNase Inhibitor, 5 mM of DTT, 1 M of Betaine, 6 mM of MgCl₂, 1 mM of dCTP (for template-switching), and 1.5 M of P5TSO. The 1st strand of cDNA was synthesized by reverse transcription at 42 °C for 90 min.

The full-length cDNAs were pre-amplified through semi-suppressive PCR for 8 or 16 cycles. The pre-amplified cDNA products were diluted and used for gene-specific PCR reactions using gene-specific primers (GSP) and poly(A) reverse primer 2. For the Ligation-Mediated Poly(A) Test (LM-PAT), oocyte samples were hybridized with oligo(dT)₂₀ at 42 °C for 30 min and then ligated with dT anchor primer by T4 DNA ligase (Sangon, B600511) at 12 °C for 2 h. Reverse transcription was performed with P5TSO and 1 mM dCTP for 1 h, ensued by PCR pre-amplification for 8 or 16 cycles using dT anchor and P5TSO. The diluted PCR products were used for gene-specific amplification. For extension, Poly(A) Test (ePAT), oocyte poly(A)-tail RNAs were 3′-prime extended using ePAT anchor primer as a template with Klenow enzyme (New England Biolabs, K0210) at 25 °C for 1 h and 80 °C for 10 min. Reverse transcription was performed with P5TSO and 1 mM dCTP for 1 h, followed by pre-amplification for 8 or 16 cycles using an ePAT anchor and P5TSO primers. PCR products were analyzed on a 2% agarose gel. All PCR primers are listed in Supplementary Data 2.

## Cell proliferation and apoptosis assays
For the cell proliferation assay, 2000 MEF and OHT-treated MEF cells were plated in 96-well plates with 200 μl of fresh complete DMEM medium supplemented with 10% FBS. After incubation for 1, 3, 5, and 7 days, cell viabilities were measured using Cell Counting Kit-8 (Med-ChemExpress, HY-K0301) following the manufacturer's instructions. For apoptosis assays, an Annexin V-FITC/PI Apoptosis Detection Kit (YEASEN, 40302ES50) was used following the manufacturer's protocol. Samples were detected by the BD Accuri C6 flow cytometry, and the results were analyzed with FlowJo V10 software.

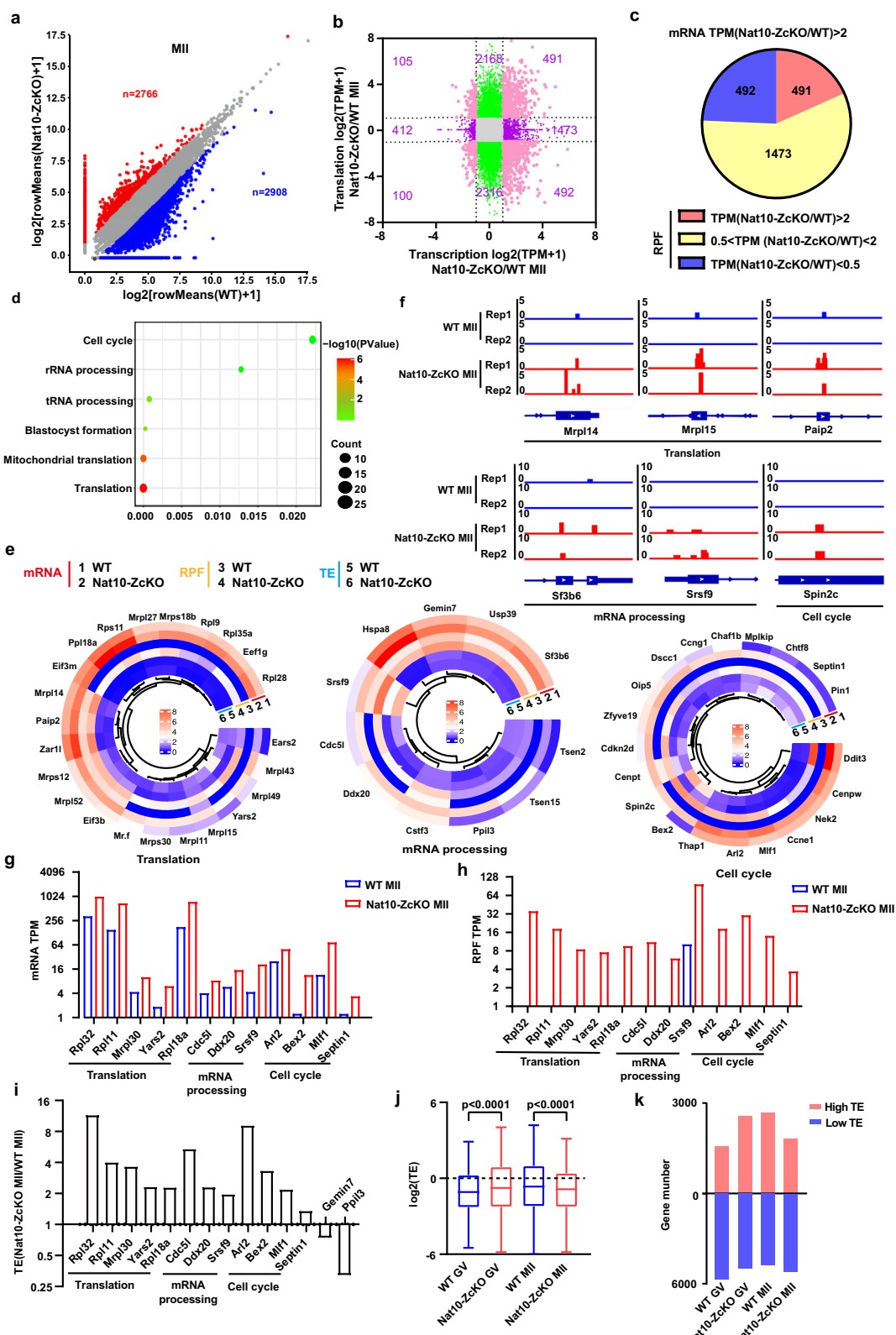

## Polysome profiling

For polysome profiling, cells were cultured in 10-cm dishes and treated with OHT for 3 days. Cells were washed with cold PBS supplemented with 100 µg/ml cycloheximide and collected by centrifugation. Cell pellets were lysed in lysis buffer [50 mM HEPES, 2 mM MgCl₂, 100 mM KCl, 100 µg/ml cycloheximide, 1 mM DTT, 0.5% Triton X-100, 10%

glycerol, and 20 U/ml EDTA-free protease inhibitor cocktail (Sigma, 11836170001)]. The lysate was cleared by centrifugation at 12,000×*g* for 10 min at 4 °C, and the supernatant was loaded onto a 20‐50% density gradient of sucrose cushion [30 mmol/l Tris−HCl (pH 7.5), 100 mmol/l NaCl, 10 mmol/l MgCl2, protease inhibitor cocktail (EDTA-free), and 100 units/ml RNase inhibitor (APExBIO, K1046)], ensued by

**Fig. 10 | Dys-regulated maternal translatome by Ribo-seq analysis in Nat10-ZcKO MII oocytes. a** Scatter plot comparing transcript RPF values calculated via average TPM between WT and Nat10-ZcKO oocytes at the MII stage. RPFs decreased or increased more than 2-fold in Nat10-ZcKO oocytes samples were highlighted in blue and red color, respectively. **b** Scatter plot comparing the gene transcript overlapping between the mRNA changes and the RPF changes in Nat10-ZcKO MII oocytes. **c** Pie chart displaying the translatome-wide distribution of maternal transcripts with up-regulated (Red), down-regulated genes (Blue) and the others (Yellow) in Nat10-ZcKO MII oocytes. **d** Gene Ontology (GO) enrichment analysis of transcript RPFs up-regulated in Nat10-ZcKO MII oocytes. **e** Heatmap of representative genes from three major functional GO categories showing up-regulated

RPFs in Nat10-ZcKO MII oocytes. **f** The UCSC browser tracks of representative genes from three major functional GO categories showing up-regulated RPF in Nat10-ZcKO MII oocytes. **g–i** Bar plots showing the relative mRNA expression levels in **g**, RPF in **h** and TE in **i** for a panel of up-regulated gene transcripts. **j** Box plots showing the relative TE values at the GV and MII stages as indicated. The box indicates the upper and lower quantiles, the thick line in the box indicates the median and the whiskers indicate 2.5th and 97.5th percentiles. Data are presented as mean ± SEM, $n = 3$. ****$p < 0.0001$ by two-tailed *Student's t-test*. **k** Bar plots showing the numbers of high-TE genes (TE > 3) and low-TE genes (TE < 0.33) in GV and MII. Source data are provided as a source data file.

ultracentrifugation in a rotor at $36,800 \times g$ for 3 h at 4 °C. After centrifugation, the gradient was fractionated and the absorbance at 254 nm was continuously recorded using an ISCO fractionator (Teledyne ISCO).

### Mini-bulk SMART-seq2 for RNA-seq library preparation

The oocyte mini-bulk SMART-seq2 protocol was based on the well-established SMART-seq2 protocol with *in-house* optimized modifications as indicated below. In brief, after the hormone challenge, five oocytes retrieved from each animal were washed at least five times in 1XPBS containing 0.5% BSA/PVP and directly lysed in Lysis Buffer MasterMix (0.3% Triton, 40 U/μl RNase inhibitor, 2.5 μM oligodT$_{30}$VN, and 2.5 μM dNTP mix). The oocyte Lysis mixture was allowed to undergo at least one freeze-thaw cycle at −80 °C to facilitate complete cytosolic lysis. The 1st strand cDNA synthesis was performed at 42 °C for 90 min, followed by 14 cycles of PCR pre-amplification to attain the full-length cDNA products through ISPCR primer-mediated semi-suppressive PCR. 1 ng of size-selected full-length cDNAs was used for Tn5-guided library preparation using Hieff NGS Fast Tagment DNA Library Prep Kit for Illumina (Yeasen, 12206ES96). Final dual-barcoded libraries were achieved through PCR amplification for 8 cycles with both index i5 and i7 primers prior to pooled library sequencing on the NovaSeq 6000 platform with PE150 mode (Novagene).

### Ribosome profiling and sequencing (Ribo-seq)

We optimized a low-input Ribo-seq protocol adapted from a ligation- and ultracentrifugation-free Ribo-seq approach[65]. In brief, -100 oocytes were collected in a pre-warmed M2 medium supplemented with cycloheximide (CHX) at a final concentration of 100 μg/ml. After incubation for 10 min, the oocytes were directly lysed in 50 μl ribosome lysis buffer (20 mM Tris–HCl at PH 7.4, 150 mM NaCl, 5 mM MgCl2, 1% Triton X-100, 1 mM DTT, 100 μg/ml CHX and 25U/ml Turbo DNase). The lysate was sheared 4× with a 26-gauge needle, and treated with 0.5 μl RNase I (100 U/μl from Invitrogen AM2294) for 1.5 h at room temperature. RNase-protected footprints were purified using TRIzol and further treated with T4 PNK (Yeasen, 12902ES86) prior to poly(A) tailing. The end-repaired RNase footprints were 3′ polyadenylated with 10 U of Escherichia coli poly(A) polymerase (Yeasen, 14801ES76) by incubating for 3 h at 37 °C. The resulting polyadenylated RNAs were reverse-transcribed by Hiscript III Reverse Transcriptase using riboRT primer and template-switching P5TSO primer at 42 °C for 90 min. The enzyme was inactivated by heating for 2 min at 85 °C. Half of the resultant cDNA product (25 μl) was combined with 25 μl of 2×Ultra II Q5 master mix, 0.5 μM index i5 and i7 primers (50 μl in total), for barcoded library PCR amplification for 18 cycles. The final DNA library was separated and visualized in a 4% agarose gel. DNA fragments with 15- to 35-bp insert bands were carefully excised and recovered using a DNA gel recovery kit, and the library sequencing was conducted on the NovaSeq 6000 platform with PE150 mode (Novagene). All PCR primers are listed in Supplementary Data 2. The Ribo-seq data in this work have been submitted to the NCBI Gene Expression Omnibus (GEO) under accession number GSE228721.

### ac4C RNA immunoprecipitation and sequencing (acRIP-seq)

We carried out a low-input acRIP-seq protocol adapted from the well-established m6A-MeRIP-seq protocol with *in-house* optimized conditions. Briefly, -2000 GV oocytes were collected and lysed with 100 μl lysis buffer (20 mM Tris–HCl at PH 7.4, 150 mM NaCl, 2 mM EDTA, 0.5% Triton X-100, 0.5% NP-40, 0.5 mM DTT, 40U murine RNase inhibitor (APExBIO, K1046), 1× Protease inhibitors and 25 U/ml Turbo DNase). The oocyte lysate was sheared 10× using a 26-gauge needle, followed by centrifugation at 4 °C with $18,000 \times g$ for 20 min and by rotating for 30 min at 4 °C. Following total RNA extraction with TRIzol, the mRNA was further purified from the total RNA using mRNA capture beads (Vazyme, R403-2). For library preparation, the mRNA was randomly fragmented to -200 nt with RNA fragmentation reagents (Thermo, AM8740), followed by incubation with the protein A beads (MedChemExpress, HY-K0203) coupled with anti-ac4C polyclonal antibody (Abcam, ab252215) or anti-IgG polyclonal antibody (Proteintech, 30000-0-AP) in acRIP buffer (150 mM NaCl, 50 mM Tris–HCl, pH 7.4, 0.05% NP-40, 0.4 U/μl RNasin) for 4 h at 4 °C. After immunoprecipitation, the RNA reaction mixture was washed twice in 1 ml of acRIP buffer, twice in 1 ml of low-salt acRIP buffer (50 mM NaCl, 50 mM Tris–HCl, pH 7.4, 0.05% NP-40, 0.4 U/μl RNasin), and twice in 1 ml of high-salt acRIP buffer (500 mM NaCl, 50 mM Tris–HCl, pH 7.4, 0.05% NP-40, 0.4 U/μl RNasin) for 5 min each at 4 °C. Next, the ac4C-enriched RNA fragments were eluted off the beads by proteinase K digestion, and were isolated by phenol–chloroform extraction and ethanol precipitation. The purified RNA fragments were subjected to library construction using the identical procedures as described in Ribo-seq protocol that incorporated end repair, 3′ polyadenylation, reverse transcription, and barcode PCR amplification. Library sequencing was performed on the NovaSeq 6000 platform with PE150 mode (Novagene).

### RNA-Seq data analysis

Raw reads were processed to remove adaptor contaminants and low-quality bases. The clean reads were aligned to the mouse genome (mm10) using STAR and uniquely mapped reads were counted with RSEM by default parameters (Supplementary Data 3). We quantified gene expression levels with TPM. For each sample, the expressed genes were defined with cutoff: TPM ≥ 1. Differentially expressed genes (DEGs) were assessed with the DESeq2 package with a cutoff: Padj <0.05 and fold change (FC) ≥ 2. Gene Ontology (GO) enrichment was performed using DAVID (https://david.ncifcrf.gov/). rMATS was used to analyze the alternative splicing events. Statistical analyses were performed using R software (http://www.rproject.org). The RNA-seq data in this work have been submitted to the NCBI Gene Expression Omnibus (GEO) under accession number SRP392832.

### Statistical analysis

All experiments were performed at least in biological triplicates unless otherwise indicated. Statistical analysis was performed using *Student's t-test* unless otherwise stated. Values of $p < 0.05$ were deemed statistically significant. Statistically significant values of $p < 0.05$, $p < 0.01$, $p < 0.001$ and $p < 0.0001$ are indicated by one, two, three and four

asterisks, respectively. Statistical data were calculated by R or Graph-Pad Prism 6.

## Reporting summary

Further information on research design is available in the Nature Portfolio Reporting Summary linked to this article.

## Data availability

The smart-seq2 in this study has been deposited in the Sequence Read Archive database under accession code SRP392832. RIP-seq and Ribo-seq data generated in this study have been deposited in the Gene Expression Omnibus database under accession code GSE228721. Source data are provided with this paper.

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

## Acknowledgements

We are grateful to Prof. Li He (School of Basic Medical Sciences, Division of Life Sciences and Medicine, University of Science and Technology of China (USTC), Hefei, Anhui) for generous assistance with oocyte imaging. We also thank all members of the Bao laboratory for the helpful discussion. This work was supported by grants from the Ministry of Science and Technology of China (2019YFA0802600 and 2022YFC2702600 to J.B.); National Natural Science Foundation of China (31970793 and 32170856 to J.B.); The open project of NHC Key Laboratory of Male Reproduction and Genetics (No. KF201901 to J.B.); the Fundamental Research Funds for the Central Universities (WK9110000181 to X.Z., WK2070000156 to J.B.) and Startup funding (KY9100000001 to J.B.) from USTC.

## Author contributions

J.B., X.Z. and W.Q. conceived, designed, and supervised the work. J.B. wrote the manuscript. X.J. performed mouse crossing, chromosome spreads analysis, immunofluorescent staining and ribo-seq and RIP-seq. Y.Ch. collected equal amounts of mouse oocytes used in this study and western blot analysis. Y.Z. did poly(A)-tail length assay. C.X. analyzed the RNA-seq data and generated the figures. X.T. and Q.L. collected mouse oocytes used for RNA-seq. X.X. did HE staining of mouse ovaries with the help of W.L. J.Z. and L.M. performed mouse genotyping. M.A. and Y.Ca. helped with the immunofluorescence.

## Competing interests

The authors declare no competing interests.
