## [Peer Review File · Nature Communications]

Maternal NAT10 orchestrates oocyte meiotic cell-cycle progression and maturation in miceEditorial Note: Parts of this Peer Review File have been redacted as indicated to maintain the confidentiality of unpublished data.

REVIEWER COMMENTS

Reviewer #1 (Remarks to the Author):

Post-transcriptional regulation of maternal RNA is the key for successful reproduction in mammals. In this study, Nat10, which catalyzes ac4C on RNA molecules, has been demonstrated to be functionally important for mouse oocyte meiotic prophase I progression, oocyte growth, and oocyte maturation. In addition, the function of Nat10 in oocyte maturation is associated with poly(A) tail length regulation, another important RNA post-transcriptional regulation. The genetic evidence here is generally convincing. The findings here are interesting and will be a good step forward to the field. This manuscript is a very good candidate for Nature communications. I have several comments for further improvement of the manuscript.

1. The localization of NAT10 in the nucleolus need to be further verified with high resolution image in SN type GV oocytes, as well as in 1-cell or 2-cell embryos.

2. How Nat10 affect NSN and SN configuration in GV stage? This is an interesting point without exploration in the current manuscript. It will be good if there are some experimental explorations. Or at least, the author shall discuss about the possible mechanisms.

3. The relative mRNA level of Cnot6l, Cnot7, and Btg4 all decreased in Nat10-ZcKO MII oocytes as shown in Fig. 8C. Early studies have shown that these three factors are with significant poly(A) tail elongation during oocyte maturation. Therefore, for the changes seen here, is it because of transcript level, or the polyadenylation status? Is this difference in mRNA level translate to protein level? In addition, how about Cnot6 and Cnot8, the other two core component of CCR4-NOT complex?

4. How NAT10 affect the level of Cnot6l, Cnot7, and Btg4? How could Nat10 mediated ac4C and translational regulation linked to the reduced mRNA level of Cnot6l, Cnot7, and Btg4?

5. Could the author provide additional evidence that the effect of Nat10-ZcKO on poly(A) tail length is mediated by the level of Cnot6l, Cnot7, and Btg4?

6. Fig. S7B is confusing, it says mRNA level in oocytes in the main text, whereas mRNA levels in ovaries in the figure legend.

7. Fig. S7 and Fig. S8 are very confusing. Since it is believed that there is no new transcription starting from GV stage. It might be reasonable to hypothesis that the maternal Nat10 shall be already there and will not be affected by TMX induced Nat10 deletion in GV oocytes in Nat10-DcKO and Nat10-UcKO as no new transcription from these KO alleles? How the explain the Nat10 level change upon TMX induced Nat10 deletion in GV oocytes of Nat10-DcKO and Nat10-UcKO? In addition, how to explain the pre-implantation development defects in Fig. S7K, L? Or do these mean that there are significant amount of new transcription of Nat10 from GV oocytes?

8. How to explain the 1-cell arrest in Nat10-UcKO in Fig. S8L? How to explain 1-cell arrest seen in Nat10-UcKO (Fig. S8L), but not seen in Nat10-ZcKO (Fig. 6H) which deletes Nat10 earlier?

9. The author mentioned that the KO MEF cells died after longer induction of KO. How long will cause the cell death? How long after induction has the experiments in Fig. 9 done? Could some of the phenotypes seen in Fig. 9 consequences of cell death?

10. There are global changes of mRNA level during oocyte maturation. Therefore, spike-ins are needed in the transcriptome analysis to better interpret the transcriptional changes in these oocyte samples as clearly demonstrated (DIO: 10.1038/nsmb.3204; 10.15252/embj.201899333). Doing transcriptome-

analysis in this way will greatly improve the accuracy of the analysis in Fig. 7.

11. HA-PAT in this study is nice. However, this assay is limited to analysis of a few pre-selected genes. Currently, there is method to analyze transcriptome-wide poly(A) tails in mammalian oocytes. I am not asking to do it in the current study. However, it will be informative to discuss about the transcriptome-wide analysis of poly(A) tails upon Nat10 KO in the oocytes in the future.

12. The breeding scheme in Fig. 4A seems to be wrong.

13. The references are not appropriate cited in many places, including but not limited to the following places:

Please add original reference for "The transcriptional activity peaks in the early growing oocytes, but gradually decreases and is considered silent in the fully grown oocyte" in the introduction.

"Interestingly, this process is accompanied by DNA configuration transition from the less condensed, non-surrounded nucleolus (NSN) to the surrounded nucleolus (SN) state, wherein fully condensed chromatin DNA encompasses the nucleolus (REF10)." The reference here is not the original reference. Please replace it with appropriate original references.

"multiple RNA modifications, such as m6A, m5C and m1A, are involved in post-transcriptional RNA metabolism, including alternative splicing, mRNA decay, and mRNA translation (REF13, 14)." The references about m5C and m1A are missing. In addition, the references for m6A are not the main ones describing the contents mentioned in the text.

14. The statement "we reasoned that Nat10 exerts its critical functions independent of ac4C acetylation" may be wrong, because Nat10 may exert its function through ac4C on rRNAs and tRNAs although not likely on mRNAs.

15. M6A has been extensively discussed in the discussion which I'd consider mostly not necessary. Please consider simplifying the discussion about m6A.

16. 1196 or 1195 genes upregulated in Nat10-ZcKO MII oocytes? There are two different numbers in the manuscript.

17. Fig. 2F and 2G are swapped between main text and figures. Please correct it.

18. Typo, "underling" in page 4.

Reviewer #2 (Remarks to the Author):

Jiang et al reported functional characterization of NAT10 during mouse oocyte maturation and early embryo development. In this study, they started with phenotype assessment and analysis using three different conditional knockouts in germ cells, targeting at meiotic stage, primary follicle stage and around antral follicles stages. At each stage, the author reported significant defects of development and attempted to reveal the underlying mechanism of the phenotypes caused by the cKO, including follicular arrest, meiotic arrest at pachytene stage, NSN-SN transition defect, and early embryo 2-cell stage arrest. RNA-seq and PolyA tail length assay were further performed to show the potential causes of the arrests and the authors concluded that NAT10 might regulate maternal RNA degradation, transcriptional and translational levels of specific genes. Although the studies presented many interesting phenotypes affecting oogenesis, but the mechanistic findings were too spread out at multiple stages of oogenesis, making the mechanistic findings a bit shallow and weak. Many of these phenotypes in germ cells have also been reported recently.

1. NAT10 have been reported to be required in many cell types as the author cited in the introduction and discussion, and the most related studies are two recent publications on oocyte maturation. (NAT10 Maintains OGA mRNA Stability Through ac4C Modification in Regulating Oocyte Maturation,

PMID: 35937804; NAT10-Mediated N4-Acetylcytidine of RNA Contributes to Post-transcriptional Regulation of Mouse Oocyte Maturation in vitro, PMID: 34395433). The published study showed knockdown of NAT10 in mouse oocyte decreases the transcript level of many genes during oocyte maturation, including OGA, leading to defects in oocyte maturation. Another report of NAT10 deletion in male germ cell caused inhibition of meiotic entry (PMID: 35801907). These studies have demonstrated the functional requirements of Nat10 for male and female germ cell development at the similar stages of the studies by Jiang et al. Thus, the novelty of the findings by Jiang et al. reduced significantly considering the published findings. Could the authors describe what are the similarity and differences of the published results with this study, especially highlighting the novelty of this study?

2. Although NAT10 expression level is relatively high in testis and ovary, the protein is expressed in many cell types. If the major enzymatic function of NAT10 is acting as ac4C writer, it may be required in all cell types and stages expressing NAT10 to regulate posttranscriptional level of the specific mRNAs. The author examined NAT10 functional requirement using 3 developmental stages of female germ cell in mice, but the gene activity may also be required since germ cell formation. Have the authors examined the expression level of NAT10 in primordial germ cell (PGCs)? Is the protein also required for PGC formation and proliferation?

3. The authors showed that Nat10-ZcKO MII oocyte might have global polyA mRNA degradation defect, and reasoned that this might be due to lower transcriptional level of CCR4-NOT subunit. Hence, it seemed that downregulation of CCR4-NOT transcription occurred before the global polyA mRNA degradation. How does NAT10 function to regulate CCR4-NOT expression? Does it regulate the expression through ac4C modification and transcript stability of CCR-NOT genes? Is it the same mechanism as the OGA finding reported in the recent study?

4. In the DcKO datasets, the authors concluded that maternal Nat10 is translationally required for pre-implantation embryo. The authors did not provide the translational profile such as Ribo-seq to demonstrate that the global level of translation was affected, instead, MEF was used to test the potential effect on translation. The cell type and developmental stage of MEF and zygote are vastly different, and there was not direct evidence of translational changes, so this part of findings and conclusion is not convincing. The author should perform translational profiling which is now available for small number or single oocyte and embryo to test their prediction.

5. The author reported that Nat10 KO cells displayed low levels of mRNA-bound polysomes than non-treated cells, therefore the translational efficiency was repressed, they further reasoned that this could be due to defect of 40S ribosomal subunit protein binding to 18s rRNA, whose ac4C modification might be perturbed by the Nat10KO. Thus, the direct defect of this phenotype might be the ac4C of 18srRNA. The authors need to provide this experimental evidence to support this conclusion. Again, the authors might have seen and demonstrated the secondary or indirect effect of Nat10 KO, whereas the primary/direct effect would be more consistent with the known enzymatic properties of the gene.

Point-by-point reply letter

We would like to thank both reviewers for the constructive comments for our manuscript. As you can see from the point-by-point reply below, following these suggestions, we have performed additional omics experiments and bioinformatic analyses. These new results were incorporated in new Fig.8, new Fig.10, new Supplementary Fig.1, new Supplementary Fig.7, new Supplementary Fig.8, new Supplementary Fig.9, as detailed below, which we believe have substantially improved this manuscript in this revised version.

Note:

Reviewers' comments=in **black** font

Our reply=in **blue** font

Body text modification=in **red** font

Response to the Referees' comments:

Reviewer #1

Comments:

Post-transcriptional regulation of maternal RNA is the key for successful reproduction in mammals. In this study, Nat10, which catalyzes ac4C on RNA molecules, has been demonstrated to be functionally important for mouse oocyte meiotic prophase I progression, oocyte growth, and oocyte maturation. In addition, the function of Nat10 in oocyte maturation is associated with poly(A) tail length regulation, another important RNA post-transcriptional regulation. The genetic evidence here is generally convincing. The findings here are interesting and will be a good step forward to the field. This manuscript is a very good candidate for Nature communications. I have several comments for further improvement of the manuscript.

Thank you for the very positive comments on this manuscript.

Question 1:

The localization of NAT10 in the nucleolus need to be further verified with high resolution image in SN type GV oocytes, as well as in 1-cell or 2-cell embryos.

Reply:

Thanks for your important comments. Following your advice, we performed high-resolution imaging by staining with the NAT10 antibody and the nucleolus-specific marker, nucleophosmin (NPM), using isolated oocytes and early embryos at various stages. As shown in Fig. R1, NAT10 is abundantly present and restricted in the nucleolus of the GO oocytes. However, NAT10 diffuses from the nucleolus to the nucleoplasm in NSN and SN oocytes, and fills in the whole cytoplasm in the MI and MII oocytes. Of note, the Nat10 signal is a bit enriched in the spindles of MI and MII oocytes owing to the nuclear breakdown. During the early pre-implantation embryonic development, the expression levels of NAT10 are reduced in 2-cell and 4-cell, but highly present in nucleolus in 8-cell, morula and blastocyst. This new data was incorporated as Supplementary Fig. 1, Page 5.

Fig. R1: The localization of NAT10 is in the nucleolus. IF images of growing oocytes (GO), non-surrounded nucleolus (NSN), surrounded nucleolus (SN), MI, MII oocytes and 2-cell, 4-cell, 8-cell, morula and blastocyst by co-staining with NAT10 antibody (Red), Nucleophosmin (NPM, Green) and DAPI (Blue) as indicated. Dashed circle indicates cellular membrane of oocytes or zona pellucida of embryos. Scale bar, 20 μ m.

Question 2:

How Nat10 affect NSN and SN configuration in GV stage? This is an interesting point without exploration in the current manuscript. It will be good if there are some experimental explorations. Or at least, the author shall discuss about the possible mechanisms.

Reply:

Thanks for your good comments. To address your concern:

1) We have found that Nat10 deletion altered the chromatin signature, causing downregulated H3K4me3 along with upregulated H3K9me3. This dys-regulated chromatin signature has been documented to closely associate with defective NSN to SN transition in ZFP36L2- and Rps26-null oocytes (Liu et al. Cell Death Dis. 2018; Jennifer N. Dumdie, Dev Cell, 2018). The levels of two key paracrine growth factors Gdf9 and Bmp15 significantly decreased in oocytes in PD14 Rps26^{fl/fl}/Gdf9-Cre mice. Insufficient expression of Gdf9 and Bmp15 in oocytes likely contributed to the arrest of oocytes growth in Rps26^{fl/fl}/Gdf9-Cre mice.

2) Furthermore, as also required by the fourth question, we performed Ribo-seq analyses using GV oocytes retrieved from WT and Nat10-ZcKO ovaries. The ribosome-protected fragments (RPFs) in Nat10-ZcKO GV oocytes were allocated into three types: Up- [TPM(Nat10-ZcKO/WT) ≥ 2], Down- [TPM(Nat10-ZcKO/WT) ≤ -2], and Stable-type (remaining RPFs). We found up-regulated RPFs in 5193 genes and down-regulated RPFs in 2501 genes in the Nat10-ZcKO GV oocytes compared to the WT oocytes (Fig. R2a). By overlapping the Smart-seq2 data with Ribo-seq result, we discovered that ~89.76% (2245/2501) of genes, which showed no change at transcript level, exhibited down-regulated RPFs enrichment in Nat10-ZcKO GV oocytes (Fig. R2b, c). Interestingly, Gene ontology (GO) analyses revealed that most RPFs-downregulated genes were enriched in cell cycle, translation and chromatin organization pathways (Fig. R2d). The levels of Gdf9 and Bmp15 significantly decreased in MII oocytes of Nat10-ZcKO mice (Fig. R2e, f). We reasoned that Nat10 ablation caused an overall translational repression although some upregulated transcripts harbor upregulated RPFs enrichment – translational enhancement.

We have now discussed the new findings in the revised version. Page 8.

Fig. R2: Nat10 is translationally required during the NSN-SN transition. **a** Scatter plot of Ribo-seq data showing differentially expressed genes (DEGs) in Nat10-ZcKO GV oocytes. Red color: Up-regulated; Blue color: Down-regulated; Cutoff: fold change [TPM (Nat10-ZcKO/WT)] ≥ 2 . **b** Scatter plot comparing the mRNA change of Nat10-ZcKO GV oocytes (Nat10-ZcKO/WT, $\log_2(\text{TPM}+1)$) (x axis) and the RPF change of Nat10-ZcKO GV oocytes (Nat10-ZcKO/WT, $\log_2(\text{TPM}+1)$) (y axis). Results are from three smart-seq2 and Ribo-seq biologically independent experiments. **c** Venn diagram showing the overlapping of transcripts that were stabilized in Nat10-ZcKO GV ($0.5 < \text{FC}[\text{TPM}(\text{Nat10-ZcKO/WT})] < 2$) and RPFs that were down-regulated in Nat10-ZcKO GV ($\text{FC}[\text{TPM}(\text{Nat10-ZcKO/WT})] < 0.5$). **d** Gene Ontology (GO) enrichment analysis of overlap genes in **c**. **e-f** Bar graph showing the RPF changing levels of Gdf9 and Bmp15 in WT and Nat10-ZcKO oocytes at the GV stages as indicated. Results shown are from merged data of three biologically independent samples.

Question 3:

The relative mRNA level of Cnot6l, Cnot7, and Btg4 all decreased in Nat10-ZcKO MII oocytes as shown in Fig. 8C. Early studies have shown that these three factors are with significant poly(A) tail elongation during oocyte maturation. Therefore, for the changes seen here, is it because of transcript level, or the polyadenylation status? Is this difference in mRNA level

translate to protein level? In addition, how about Cnot6 and Cnot8, the other two core component of CCR4-NOT complex?

Reply:

Thanks for your good advice. Following your suggestion, we collected more MII oocytes and re-analyzed the mRNA expression levels of Cnot6l, Cnot7 and Btg4 in WT and KO MII oocytes by qPCR. We found that the mRNA levels of Cnot7 and Btg4 were significantly decreased. The Cnot6l mRNA levels were also reduced but without statistically significant difference between the WT and KO MII oocytes (Fig. R3a). Interestingly, the mRNA levels for Cnot6 and Cnot8 increased likely through a compensatory mechanism when Cnot7 and Btg4 levels were declined (Fig. R3b). In addition, HA-PAT and LM-PAT were used to analyze the poly(A) tail length of Cnot6l, Cnot7 and Btg4, and the results showed that the expression levels of these three genes were not related to the poly(A) tail length (Fig. R3c, d). Strikingly, further immunoblotting analyses demonstrated that the protein levels for CNOT6L, CNOT7 and BTG4 were vastly declined in MII oocytes from Nat10-ZcKO mice (Fig. R3e). This evidence suggest that Nat10 has an impact on both the transcriptional and translational regulation of CNOT6L, CNOT7 and BTG4 in MII oocytes. These new data have been incorporated into new Fig. 8. in this revised version. Page 10.

Fig. R3: Nat10 modulates the expression of CNOT6L, CNOT7 and BTG4 at both the transcriptional and translational levels in MII oocytes from Nat10-ZcKO mice. a-b qPCR results showing the relative levels of indicated transcripts (Cnot6l, Cnot7 and Btg4) in a and (Cnot6 and Cnot8) in b in WT and Nat10-ZcKO oocytes at MII stage. Data are presented as the mean±SEM, n=3. n.s., non-significant, *, p<0.05, **, p<0.01, ***, p<0.001 by two-tailed *Student's t-test*. c-d HA-PAT assay in c and LM-PAT in d results showing changes in poly(A)-tail lengths of indicated transcripts in WT and Nat10-ZcKO oocytes at GV and MII stages. Experiments were performed in triplicates; a representative image is shown in the 2% agarose gel (Left) and the length distribution shown in the densitometric curves (Right). e Western blot displaying the NAT10, CNOT6L, CNOT7 and BTG4 protein levels in MII oocytes of WT and Nat10-ZcKO mice. α-TUBULIN was used as a loading control.

Question 4:

How NAT10 affect the level of Cnot6l, Cnot7, and Btg4? How could Nat10 mediated ac4C and translational regulation linked to the reduced mRNA level of Cnot6l, Cnot7, and Btg4?

Reply:

Thanks for your good comments. Following your advice, we first performed acRIP-seq (as adopted by Arango D et al. Cell. 2018) with pools of 2000 WT GV oocytes. Gene ontology (GO) analyses revealed that the transcripts with ac4C modification were mostly enriched in cell cycle and mRNA processing pathways (Fig. R4a). We next verified the validity of our acRIP-seq data by RIP-PCR for genes which have been shown to harbor ac4C modification in HeLa cells published (Arango D et al. Cell. 2018 Dec 13;175(7):1872-1886.e24). We selected Bst2, Fus and Nomo as the positive control (ac4C+), and Eef1a1 and Gapdh as the negative control (ac4C-). Our method clearly detected the bands for ac4C+ gene transcripts with ac4C antibody but not IgG antibody. For ac4C- transcripts as negative controls, there is no band seen in either ac4C or IgG beads. Moreover, the band intensity in the ac4C supernatant for ac4C+ genes were also present but with reduced levels, suggesting the pulldown efficiency by the ac4C antibody binding was not 100% efficient (Fig. R4b). This result confirmed the validity and sensitivity of our acRIP data. Next, we performed RIP-PCR to verify whether Cnot6l, Cnot7 and Btg4 harbor ac4C modification under the same condition. This assay revealed that all three gene transcripts were clearly modified by ac4C, albeit with reduced signal seen for Cnot6l as compared with Cnot7 and Btg4 (Fig. R4c, d), further verifying the validity of our acRIP-seq data. These results suggest that Nat10 might promote the stability of Cnot7 and Btg4 mRNA by catalyzing ac4C modification.

To further address whether Nat10 is involved in translational regulation, we performed Ribo-seq using GV and MII oocytes collected from WT and Nat10-ZcKO mice by optimizing a protocol for low-input samples (Li et al. Genome Res. 2022 Mar;32(3):545-557). We finally set up a protocol for Ribo-seq using only 100 oocytes, which utilizes ligation-free library preparation but does not rely on ultracentrifugation. Compared with the published data by Sha et al. using 500 oocytes (Sha et al. EMBO J. 2018 Dec 14;37(24):e99333), our Ribo-seq protocol detected comparable numbers of translated genes and global translational patterns (Fig. R5a-c). A snapshot of the IGV browser tracks displayed characteristic RPF (ribosome-protected fragments) signatures at the exon regions for randomly selected genes by both approaches (Fig. R5c). On average, our method detected

~9425 genes in GV and MII oocytes (Fig. R5d). The mapped reads showed high coverage in the coding sequence (CDS) and low coverage in the 3'UTRs (Fig. R5e). The 3-nt periodicity (Fig. R5f) and good correlations were also observed in all samples (Fig. R5g).

Ribosome profiling data unveiled that the RPFs for a total of 2619 transcripts were up-regulated while 3632 transcript RPFs down-regulated in Nat10-ZcKO MII oocytes (Fig. R6a). We further overlapped the RPF translome and the transcriptome of MII oocytes, and found that 491 gene transcripts significantly accumulated in Nat10-ZcKO MII oocytes displayed up-regulated RPF enrichment in translome (Fig. R6b, c). GO analyses revealed that those gene transcripts were predominantly enriched in translation, RNA processing and blastocyst formation (Fig. R6d-f). In addition, strikingly, the majority of the genes displaying the up-regulated levels in both translome and transcriptome in Nat10-ZcKO MII oocytes exhibited increased TE (Translational Efficiency=RPF/mRNA) (Fig. R6g-i). However, it is worth to note that the overall TE levels decreased in Nat10-ZcKO MII oocytes, suggesting Nat10 loss caused translational repression in MII oocytes (Fig. R6j-k). These new data have been incorporated into new Fig. 10. Supplementary Fig.7,8,9 in this revised version. Page 12-13.

Fig. R4: Nat10 deposits ac4C modification on the key transcripts from CCR4-NOT complex genes. **a** Gene Ontology (GO) enrichment analysis of gene transcripts with ac4C modification identified by RIP-seq in the GV oocytes. **b** acRIP-PCR assay confirmed selected genes with or without ac4C modification in HeLa cells. IgG B, IgG beads. Ac4C B, ac4C beads. IgG S, IgG supernatant. Ac4C S, ac4C supernatant. Experiments were performed in triplicates; a representative image is shown in the 2% agarose gel. **c** acRIP-PCR results validated the ac4C modification on the transcripts for Cnot6l, Cnot7 and Btg4 in WT GV oocytes. **d** acRIP-qPCR assay showing the efficiency of IP for Cnot6l, Cnot7 and Btg4 transcripts in WT GV oocytes. Data are presented as the mean \pm SEM, $n=3$; ****, $p<0.0001$ by two-tailed *Student's t-test*.

Fig. R5: Genome-wide profiling of translome by Ribo-seq in the oocytes. **a** Bar plots showing the numbers of detected genes (TPM>1) by the published Ribo-seq data of Sha et al. (two replicates, 'Rep1' and 'Rep2') and

Ribo-seq in this study (three replicates, 'Rep1', 'Rep2' and 'Rep3'). **b** Scatter plots comparing the Ribo-seq data from Sha et al. and this study. Spearman correlations are also shown. **c** The UCSC browser tracks of Ribo-seq signal (RPF) for Zp2, Tubb2a and Cpeb1. **d** Bar plots showing the numbers of genes detected by Ribo-lite (TPM > 1). **e** Bar plots showing the mapping reads distribution in 5' UTR, CDS and 3'UTR regions. **f** Bar plots showing the percentages of footprints that match the reading frames. **g** Scatter plots comparing replicates of Ribo-seq for GV and MII from WT and Nat10-ZcKO mice.

Fig. R6: Dys-regulated maternal translome by Ribo-seq analysis in Nat0-ZcKO MII oocytes.

a Scatter plot comparing transcript RPF values calculated via average TPM between WT and Nat10-ZcKO oocytes at MII stage. RPFs decreased or increased more than 2-fold in Nat10-ZcKO oocytes samples were highlighted in blue and red color, respectively. **b** Scatter plot comparing the gene transcript overlapping between the mRNA changes and the RPF changes in Nat10-ZcKO MII oocytes. **c** Pie chart displaying the translome-wide distribution of maternal transcripts with up-regulated (Red), down-regulated genes (Blue) and the others (Yellow) in Nat10-ZcKO MII oocytes. **d** Gene Ontology (GO) enrichment analysis of transcript RPFs up-regulated in Nat10-ZcKO MII oocytes. **e** Heatmap of representative genes from three major functional GO categories showing up-regulated RPFs in Nat10-ZcKO MII oocytes. The color intensity gradient from red to blue indicates the relative gene expression levels from high to low. **f** The UCSC browser views of representative genes from three major functional GO categories showing up-regulated RPF in Nat10-ZcKO MII oocytes. **g-i** Bar plots showing the relative mRNA expression levels in **g**, RPF in **h** and TE in **i** for a panel of up-regulated gene transcripts. **j** Box plots showing the relative TE values at the GV and MII stages as indicated. Data are presented as mean \pm SEM, n=3. ****, p<0.0001 by two-tailed *Student's t-test*. **k** Bar plots showing the numbers of high-TE genes (TE >3) and low-TE genes (TE <0.33) in GV and MII.

Question 5:

Could the author provide additional evidence that the effect of Nat10-ZcKO on poly(A) tail length is mediated by the level of Cnot6l, Cnot7, and Btg4?

Reply:

Thanks for your comments. We carried out western blot analysis demonstrating that the protein levels for CNOT6L, CNOT7 and BTG4 were vastly declined in MII oocytes from Nat10-ZcKO compared with WT mice (Fig. R3e). see new Fig.8.

Question 6:

Fig. S7B is confusing, it says mRNA level in oocytes in the main text, whereas mRNA levels in ovaries in the figure legend.

Reply:

Thanks for your comments. We have removed Figure S7, see question 7 below.

Question 7:

Fig. S7 and Fig. S8 are very confusing. Since it is believed that there is no new transcription starting from GV stage. It might be reasonable to hypothesis that the maternal Nat10 shall be already there and will not be affected by TMX induced Nat10 deletion in GV oocytes in Nat10-DcKO and Nat10-UcKO as no new transcription from these KO alleles? How the explain the Nat10 level change upon TMX induced Nat10 deletion in GV oocytes of Nat10-DcKO and Nat10-UcKO? In addition, how to explain the pre-implantation development defects in Fig. S7K, L? Or do these mean that there are significant amount of new transcription of Nat10 from GV oocytes?

Reply:

Thanks for your comments. We performed the TMX mock injection in the WT females (WT-TMX) as an independent group, and repeated the same injections for other two groups. We found that in both the WT-TMX and Nat10-DcKO groups, the Nat10 mRNA levels were decreased by ~20% in the Nat10 cKO ovaries as compared with those in WT mice without injection (Fig. R7a, d). We speculated that this reduction was likely ascribed to a small number of growing oocytes which harbor active transcription of Nat10.

After more biological repeats and careful statistical calculations, we found that the TMX injection itself can cause the unwanted side-effects in the ovaries on pre-implantation embryonic development (Fig. R7c, f), making the WT-TMX group significantly distinguishable from the WT group. We are now exploring this effect (to be an independent project since it connects to the pre-implantation embryonic development) and preferred to remove both Figure S7 and Figure S8 for clarification. The main body texts have also been removed and replaced by the mechanistic omics data in this revised version of MS. Page 12-13

[redacted]

Question 8:

How to explain the 1-cell arrest in Nat10-UcKO in Fig. S8L? How to explain 1-cell arrest seen in Nat10-UcKO (Fig. S8L), but not seen in Nat10-ZcKO (Fig. 6H) which deletes Nat10 earlier?

Reply:

Thanks for your comments. As demonstrated above, we have carried out additional TMX mock injection in the WT mice and confirmed the phenotypic side-effects in the WT mice by the injection of TMX itself during the pre-implantation embryonic development as noted above (Fig. R7d-f). We reasoned that this side-effect toxicity by TMX injection could be ascribed to the functional Estrogen receptor (ER)-mediated signaling present in the late stages of follicle development. Therefore, we removed both original Figure S7 and Figure S8 for clarification.

Question 9:

The author mentioned that the KO MEF cells died after longer induction of KO. How long will cause the cell death? How long after induction has the experiments in Fig. 9 done? Could some of the phenotypes seen in Fig. 9 consequences of cell death?

Reply:

Thanks for your comments. Following OHT treatment for three consecutive days, apparent cell death could be observed starting from Day 5 post-OHT treatment and almost all cells died at Day 10 post-OHT treatment. The results shown in Fig. 9 were achieved at Day 2 after OHT treatment for three consecutive days, when no cell death or cell apoptosis was observed.

Question 10:

There are global changes of mRNA level during oocyte maturation. Therefore, spike-ins are needed in the transcriptome analyses to better interpret the transcriptional changes in these oocyte samples as clearly demonstrated (DIO: 10.1038/nsmb.3204; 10.15252/embj.201899333). Doing transcriptome-analyses in this way will greatly improve the accuracy of the analyses in Fig. 7.

Reply:

Thanks for your suggestion. We do appreciate the potential advantages that come with inclusion of ERCC spike-in into the RNA-seq pipeline analysis. As a matter of fact, at the beginning of this study, there was an out of stock for ERCC and it is considerably expensive (2297 \$/10 μ L), we thus chose not to include the spike-in. Instead, we have put much more efforts in optimizing the mini-Smart-seq2 protocol (which has indeed been updated to Smart-seq3 recently, Michael Hagemann-Jensen, et al, Nature Biotechnology, 2020) specifically for oocyte transcriptome analysis (see methods), and indeed, after normalization, our optimized mini-Smart-seq2 pipeline was more sensitive and has detected more gene numbers (~12000 gene, TPM>1) in the single GV oocytes, as compared with the published papers (~11200) (DIO: 10.1038/nsmb.3204; 10.15252/embj.201899333) (Fig. R8e).

On the other hand, since the overarching goal to solve this reviewer's concern is to improve accuracy of transcriptome analysis, we next compared the RNA-seq data with (ERCC) or without (standard) normalization by ERCC using oocyte (GV and MII) data available from Sha et al. (Sha et al. EMBO J. 2018 Dec 14), and further compared their ERCC data with our results as detailed below:

First, we compared the gene numbers detected in either Standard or ERCC-normalized group at GV and MII stages for WT and KO oocytes. This unveiled that, in general, the average numbers of detectable genes (TPM>1) were comparable between Standard and ERCC-normalized groups, with approximately 14-17 more genes detected in ERCC-normalized group (Fig. R8a). We next examined the differentially expressed genes (DEGs) by running RNA-seq pipeline (either htseq or rsem), and found that there was a high

overlapping rate between the standard and ERCC groups (>90% overall) (Fig. R8b).

Next, the Principal component analyses (PCA) of read counts after normalization also showed clear intra-sample clustering between the standard and ERCC groups, while there was a clear separation among different samples (Fig. R8c, d). Overall, in WT MII oocytes, the average number of reads detected in our smart-seq2 data was higher than that of sha et al. in standard or normalized ERCC groups (Fig. R8e). Additionally, after removing the batch effect, the PCA showed that there was no distinction in the transcriptome of WT MII oocytes between our data and their normalized ERCC data (Fig. R8f). When comparing the “DEGs” in WT MII oocytes between our data and the normalized ERCC group, only a total of 25 genes (24 up-regulated vs 1 down-regulated) popped out by htseq pipeline, whereas 8 genes (3 down-regulated vs 5 up-regulated) were considered as “DEGs” by rsem calculation (Fig. R8g), signifying that the improvement by ERCC normalization is marginal for our optimized protocol. Altogether, these data suggest that our optimized Smart-seq2 protocol was more sensitive, and the ERCC normalization does improve the accuracy but with limited improvement. Given the enormous cost for the sequencing and the great efforts we have put into the optimization of Smart-seq2 approach, we prefer to keep this consistent data.

Fig. R8: Comparison among Smart-seq2 datasets with and without ERCC normalization. **a** Bar plots showing the numbers of detected genes (TPM >1) by Smart-seq2 data normalized with (ERCC) or without ERCC (Standard) from Sha et al. (Sha et al. EMBO J. 2018). n.s., non-significant by two-tailed *Student's t-test*. **b** Venn diagram showing the overlapping of DEGs, normalized by ERCC and Standard, after counting by htseq (Top) or rsem (Bottom) in KO GV or MII. **c** PCA plot of smart-seq2 sequencing data normalized by ERCC or Standard and counted by htseq or rsem in GV or MII of WT and KO mice. **d** Heatmap of smart-seq2 sequencing data normalized by ERCC or Standard and counted by htseq or rsem in GV or MII of WT and KO mice. **e** Bar plots showing the numbers of detected genes (TPM >1) by this study smart-seq2 data and reference smart-seq2 data normalized with (ERCC) or without ERCC (Standard) in WT MII oocytes. **f** PCA plot of reference smart-seq2 data normalized with ERCC and this study smart-seq2 data, after removing the batch effect in WT MII oocytes. **g** Heatmap of the DEGs between this study smart-seq2 sequencing data and reference smart-seq2 data normalized by ERCC after counting by htseq (Top) or rsem (Bottom) in WT MII oocytes.

Question 11:

HA-PAT in this study is nice. However, this assay is limited to analyses of a few pre-selected genes. Currently, there is method to analyze transcriptome-wide poly(A) tails in mammalian oocytes. I am not asking to do it in the current study. However, it will be informative to discuss about the transcriptome-wide analyses of poly(A) tails upon Nat10 KO in the oocytes in the future.

Reply:

We have discussed this as suggested, Page 15.

Question 12:

The breeding scheme in Fig. 4A seems to be wrong.

Reply:

Changed as suggested on Fig. 4A.

Question 13:

The references are not appropriate cited in many places, including but not limited to the following places:

Please add original reference for “The transcriptional activity peaks in the early growing oocytes, but gradually decreases and is considered silent in the fully grown oocyte” in the introduction.

“Interestingly, this process is accompanied by DNA configuration transition from the less condensed, non-surrounded nucleolus (NSN) to the surrounded nucleolus (SN) state, wherein fully condensed chromatin DNA encompasses the nucleolus (REF10).” The reference here is not the original reference. Please replace it with appropriate original references.

“multiple RNA modifications, such as m6A, m5C and m1A, are involved in post-transcriptional RNA metabolism, including alternative splicing, mRNA decay, and mRNA translation (REF13, 14).” The references about m5C and m1A are missing. In addition, the references for m6A are not the main ones describing the contents mentioned in the text.

Reply:

Thanks for your comments. We have gone through the body texts and added the references as suggested.

Question 14:

The statement “we reasoned that Nat10 exerts its critical functions independent of ac4C acetylation” might be wrong, because Nat10 might exert its function through ac4C on rRNAs and tRNAs although not likely on mRNAs.

Reply:

Thanks for your comments. Changed as suggested.

Question 15:

M6A has been extensive discussed in the discussion which I’d consider mostly not necessary. Please consider simplifying the discussion about m6A.

Reply:

Thanks for your advice. We have removed some discussion texts as suggested.

Question 16:

1196 or 1195 genes upregulated in Nat10-ZcKO MII oocytes? There are two different numbers in the manuscript.

Reply:

Changed to 1196 as suggested.

Question 17:

Fig. 2F and 2G are swapped between main text and figures. Please correct it.

Reply:

Changed as suggested.

Question 18:

Typo, “underling” in page 4.

Reply:

Changed as suggested.

Reviewer #2

Comments:

Jiang et al reported functional characterization of NAT10 during mouse oocyte maturation and early embryo development. In this study, they started with phenotype assessment and analyses using three different conditional knockouts in germ cells, targeting at meiotic stage, primary follicle stage and around antral follicles stages. At each stage, the author reported significant defects of development and attempted to reveal the underlying mechanism of the phenotypes caused by the cKO, including follicular arrest, meiotic arrest at pachytene stage, NSN-SN transition defect, and early embryo 2-cell stage arrest. Smart-seq2 and PolyA tail length assay were further performed to show the potential causes of the arrests and the authors concluded that NAT10 might regulate maternal RNA degradation, transcriptional and translational levels of specific genes. Although the studies presented many interesting phenotypes affecting oogenesis, but the mechanistic findings were too spread out at multiple stages of oogenesis, making the mechanistic findings a bit shallow and weak. Many of these phenotypes in germ cells have also been reported recently.

Thank you for these comments on this manuscript. As you can see in this revised version,

we have spent great efforts, and comprehensively performed the RIP-seq and Ribo-seq omics data analyses, which were not trivial but extremely challenging when working with low-input samples, e.g., using oocytes. Please see the replies as detailed below.

Question 1:

NAT10 have been reported to be required in many cell types as the author cited in the introduction and discussion, and the most related studies are two recent publications on oocyte maturations. (NAT10 Maintains OGA mRNA Stability Through ac4C Modification in Regulating Oocyte Maturation, PMID: 35937804; NAT10-Mediated N4-Acetylcytidine of RNA Contributes to Post-transcriptional Regulation of Mouse Oocyte Maturation in vitro, PMID: 34395433). The published study showed knockdown of NAT10 in mouse oocyte decreases the transcript level of many genes during oocyte maturation, including OGA, leading to defects in oocyte maturation. Another report of NAT10 deletion in male germ cell caused inhibition of meiotic entry (PMID: 35801907). These studies have demonstrated the functional requirements of Nat10 for male and female germ cell development at the similar stages of the studies by Jiang et al. Thus, the novelty of the findings by Jiang et al. reduced significantly considering the published findings. Could the authors describe what are the similarity and differences of the published results with this study, especially highlighting the novelty of this study?

Reply:

Thanks for your comments. We noticed that there are two papers related to Nat10 function in oocyte development recently published during the preparation of this ongoing project (NAT10 Maintains OGA mRNA Stability Through ac4C Modification in Regulating Oocyte Maturation, PMID: 35937804; NAT10-Mediated N4-Acetylcytidine of RNA Contributes to Post-transcriptional Regulation of Mouse Oocyte Maturation in vitro, PMID: 34395433). However, there are major inevitable caveats associated with both two papers:

1) The conclusions from both published papers were solely based on *in-vitro* studies by oocyte microinjection (Nat10 siRNA knockdown or Trim-away *in vitro*), without any animal genetic evidence showing the necessity of Nat10 function through oocyte development. By comparison, in this study we have utilized four different mouse models that exhaustively

dissected the genetic requirement *in vivo* for Nat10 roles at varied stages of oocyte development;

2) There are no global studies using either Ribo-seq or RIP-seq to explore the Nat10 mRNA targets in both published papers. Even for the OGA transcript, there is lacking of evidence showing the ac4C-mediated transcript stability in those *in vitro* studies. Furthermore, the knockdown of Nat10 and OGA genes caused opposite dysregulation of the global transcriptome, suggesting OGA transcript de-stability in the absence of Nat10 is unlikely to be the direct cause of oocyte maturation arrest. In this revised study, we have now performed the global Ribosome profiling and RIP sequencing to identify the direct effect of Nat10 deletion. We showed that the transcripts (Cnot6l, Cnot7 and Btg4) from CCR4-NOT complex clearly harbor the ac4C modification. Upon Nat10 ablation, the mRNA expression levels and in particular the protein levels were significantly declined, causing the defective degradation and accumulation of polyA-containing transcripts responsible for oocyte meiotic arrest.

3) Both papers only performed RNA-seq in GV oocytes, while we conducted the mini-bulk SMART-seq2 using GV and MII oocytes in mouse models. We found the maternal mRNAs decay was defective at MII stage. we used a novel *in-house* devised method, termed Hairpin Adaptor-Poly(A) Tail length (HA-PAT), which is more sensitive, cost-efficient and time-saving than previous methods, and suggested that Nat10 sculpted the transcriptome of MII oocytes by poly(A)-tail shortening.

4) We conducted global Ribo-seq of GV and MII oocytes in WT and Nat10-ZcKO mice. The aberrantly accumulated transcripts in Nat10-ZcKO oocytes, which were enriched in translation and mRNA processing-related biological processes, possess up-regulated RPF enrichment in the Nat10-ZcKO MII oocytes;

5) Lin et al. overlapped the down-regulated transcripts in Nat10 knockdown GV oocytes with the acRIP-seq data from HeLa cells (Arango D et al. Cell. 2018 Dec 13;175(7):1872-1886.e24), thus indirectly inferred the potential ac4C sites in a total of 18 DEGs, among which the abundance of OGA was high (>1). As mentioned by this reviewer, the somatic cells are different from oocytes, thus this conclusion is not reliable in the absence of additional experimental validation. We conducted acRIP-seq of WT GV with pools of ~2,000 oocytes. Also, we jointly analyzed the omics data from Smart-seq2, Ribo-seq and acRIP-seq, and

provided the functional link of ac4C modification to mRNA degradation and translation in oocytes.

6) The role of Nat10 in male germline development is a different story that is not covered by this female MS. Although it was published recently in NAR, its mechanism is still largely unclear. We are also working this project as an independent project.

Question 2:

Although NAT10 expression level is relatively high in testis and ovary, the protein is expressed in many cell types. If the major enzymatic function of NAT10 is acting as ac4C writer, it might be required in all cell types and stages expressing NAT10 to regulate posttranscriptional level of the specific mRNAs. The author examined NAT10 functional requirement using 3 developmental stages of female germ cell in mice, but the gene activity might also be required since germ cell formation. Have the authors examined the expression level of NAT10 in primordial germ cell (PGCs)? Is the protein also required for PGC formation and proliferation?

Reply:

Thanks for your comments. We assessed the single-cell RNA-seq data published by Zhao et al. (Zhao et al. Nat Commun. 2021 Nov 25;12(1):6839), and found that the mRNA expression levels of Nat10 in PGCs are slightly higher than those in meiotic spermatocytes, but lower than those in spermatogonia (Fig. R9a). To explore the effects of Nat10 on the formation of PGCs, we first generated the *Nat10^{lox/-}* mice and next performed the crossing between the male and female *Nat10^{lox/-}* mice. However, we found that there is a clear pre-implantation embryonic arrest prior to Embryonic day 7 in *Nat10^{-/-}* embryos, which hindered us from dissecting the PGC for further functional analyses. On the other hand, since the PGC formation starts from the epiblast of the inner cell mass, by utilizing the unique Ubc-cre; *Nat10^{lox/lox}* mouse model, we successfully derived and established two ESC cell lines, wherein the TMX injection will cause the Nat10 deletion in the ESC cells.

After OHT treatment for three consecutive days, Nat10 protein was fully abrogated as

evidenced by the western blot analysis (Fig. R9b). We found that, the numbers and the sizes of ESC clones significantly decreased owing to the loss of Nat10 (Fig. R9c, d), and there is also a marked reduction in cell proliferation, as examined by CCK8 kit (Fig. R9e). Moreover, by immunofluorescence staining with the stem cell marker Nanog, we found that the expression levels of Nanog decreased after Nat10 knockout. These data suggest that Nat10 is essential to maintain the self-renewal and proliferation of ESCs (Fig. R9f). However, since this study only covers the oocyte meiotic progression subsequent to PGC formation, as stated in the manuscript title “Maternal N-acetyltransferase 10 (NAT10) orchestrates oocyte meiotic cell-cycle progression and maturation in mice”, thus further exploration of Nat10 function during PGC formation is beyond the scope of this study, and is an independent, ongoing project in our lab.

[redacted]

Question 3:

The authors showed that Nat10-ZcKO MII oocyte might have global polyA mRNA degradation

defect, and reasoned that this might be due to lower transcriptional level of CCR4-NOT subunit. Hence, it seemed that downregulation of CCR4-NOT transcription occurred before the global polyA mRNA degradation. How does NAT10 function to regulate CCR4-NOT expression? Does it regulate the expression through ac4C modification and transcript stability of CCR-NOT genes? Is it the same mechanism as the OGA finding reported in the recent study?

Reply:

Thanks for your good advice. Following your suggestion, we collected more MII oocytes and re-analyzed the mRNA expression levels of Cnot6l, Cnot7 and Btg4 in WT and KO MII oocytes by qPCR. We found that the mRNA levels of Cnot7 and Btg4 were significantly decreased. The Cnot6l mRNA levels were also reduced but without statistically significant difference between the WT and KO MII oocytes (Fig. R3a). Interestingly, the mRNA levels for Cnot6 and Cnot8 increased likely through a compensatory mechanism when Cnot7 and Btg4 levels were declined (Fig. R3b). In addition, HA-PAT and LM-PAT were used to analyze the poly(A) tail length of Cnot6l, Cnot7 and Btg4, and the results showed that the expression levels of these three genes were not related to the poly(A) tail length (Fig. R3c, d). Strikingly, further immunoblotting analyses demonstrated that the protein levels for CNOT6L, CNOT7 and BTG4 were vastly declined in MII oocytes from Nat10-ZcKO mice (Fig. R3e). This evidence suggest that Nat10 has an impact on both the transcriptional and translational regulation of CNOT6L, CNOT7 and BTG4 in MII oocytes. These new data have been incorporated into new Fig. 8. in this revised version. Page 43.

We next performed acRIP-seq (as adopted by Arango D et al. Cell. 2018) by optimization with pools of >1,000 WT GV oocytes. Gene ontology (GO) analyses revealed that the transcripts with ac4C modification were mostly enriched in cell cycle and mRNA processing pathways (Fig. R4a). We subsequently verified the validity of our acRIP-seq data by RIP-PCR for genes which have been shown to harbor ac4C modification in Hela cells published (Arango D et al. Cell. 2018 Dec 13;175(7):1872-1886.e24). We selected Bst2, Fus and Nomo as the positive control (ac4C+), and Eef1a1 and Gapdh as the negative control (ac4C-). Our method clearly detected the bands for ac4C+ gene transcripts with ac4C antibody but not

IgG antibody. For ac4C- transcripts as negative controls, there is no band seen in either ac4C or IgG beads. Moreover, the band intensity in the ac4C supernatant for ac4C+ genes were also present but with reduced levels, suggesting the pulldown efficiency by the ac4C antibody binding was not 100% efficient (Fig. R4b). This result confirmed the validity and sensitivity of our acRIP data. Next, we performed RIP-PCR to verify whether Cnot6l, Cnot7 and Btg4 harbor ac4C modification under the same condition. This assay revealed that all three gene transcripts were clearly modified by ac4C, albeit with reduced signal seen for Cnot6l as compared with Cnot7 and Btg4 (Fig. R4c, d), further verifying the validity of our acRIP-seq data. These results suggest that Nat10 might promote the stability of Cnot7 and Btg4 mRNA by catalyzing ac4C modification. (Page 12-13).

Question 4:

In the DcKO datasets, the authors concluded that maternal Nat10 is translationally required for pre-implantation embryo. The authors did not provide the translational profile such as Ribo-seq to demonstrate that the global level of translation was affected, instead, MEF was used to test the potential effect on translation. The cell type and developmental stage of MEF and zygote are vastly difference, and there was not direct evidence of translational changes, so this part of findings and conclusion is not convincing. The author should perform translome profiling which is now available for small number or single oocyte and embryo to test their prediction.

Reply:

Thanks for your good advice. To further address whether Nat10 is involved in translational regulation, we repeatedly tried to optimize and performed Ribosome profiling sequencing using GV and MII oocytes collected from WT and Nat10-ZcKO mice by adapting a protocol for low-input samples (Li et al. Genome Res. 2022 Mar;32(3):545-557). We finally set up a protocol for Ribo-seq using only 100 oocytes, which utilizes ligation-free library preparation but does not rely on ultracentrifugation. Thus, this Ribo-seq approach is more affordable with less loss of input materials. Compared with the published data by Sha

et al. using 500 oocytes (Sha et al. EMBO J. 2018 Dec 14;37(24):e99333), our Ribo-seq protocol detected comparable numbers of translated genes and global translational patterns (Fig. R5a-c). A snapshot of the IGV browser tracks displayed characteristic RPF (ribosome-protected fragments) signatures at the exon regions for randomly selected genes by both approaches (Fig. R5c). On average, our method detected ~9425 genes in GV and MII oocytes (Fig. R5d). The mapped reads showed high coverage in the coding sequence (CDS) and low coverage in the 3'UTRs (Fig. R5e). The 3-nt periodicity (Fig. R5f) and good correlations were also observed in all samples (Fig. R5g). (Page 12-13)

Ribosome profiling data unveiled that the RPFs for a total of 2619 transcripts were up-regulated while 3632 transcript RPFs down-regulated in Nat10-ZcKO MII oocytes (Fig. R6a). We further overlapped the RPF translome and the transcriptome of MII oocytes, and found that 491 gene transcripts significantly accumulated in Nat10-ZcKO MII oocytes displayed up-regulated RPF enrichment in translome (Fig. R6b, c). GO analyses revealed that those gene transcripts were predominantly enriched in translation, RNA processing and blastocyst formation (Fig. R6d-f). In addition, strikingly, the majority of the genes displaying the up-regulated levels in both translome and transcriptome in Nat10-ZcKO MII oocytes exhibited increased TE (Translational Efficiency=RPF/mRNA) (Fig. R6g-i). However, it is worth to note that the overall TE levels decreased in Nat10-ZcKO MII oocytes, suggesting Nat10 loss caused translational repression in MII oocytes (Fig. R6j-k). (Page 12-13)

Question 5:

The author reported that Nat10 KO cells displayed low levels of mRNA-bound polysomes than non-treated cells, therefore the translational efficiency was repressed, they further reasoned that this could be due to defect of 40S ribosomal subunit protein binding to 18s rRNA, whose ac4C modification might be perturbed by the Nat10KO. Thus, the direct defect of this phenotype might be the ac4C of 18srRNA. The authors need to provide this experimental evidence to support this conclusion. Again, the authors might have seen and

demonstrated the secondary or indirect effect of Nat10 KO, whereas the primary/direct effect would be more consistent with the known enzymatic properties of the gene.

Reply:

Thanks for your comments. we have removed the rRNA modification data. To explore the primary/direct effect, we have comprehensively performed the RIP-seq and Ribo-seq using mouse oocytes, and jointly analyzed the omics datasets from Smart-seq2, RIP-seq and Ribo-seq, as shown above in Question 4 (Fig.R10). All the new omics experiments and data have now been incorporated into new Fig. 10. Supplementary Fig.7,8,9 in this revised version. Page 12-13.

Fig. R10: ac4C modification regulates the translation efficiency for transcripts enriched in translation, chromosome segregation and tRNA processing. **a** Venn diagram showing the overlapping of transcripts with ac4C modification (FC [TPM (ac4C/IgG)]>2), up-regulated RPF (FC [TPM (Nat10-ZcKO/WT)]>2) and up-regulated mRNA (FC[TPM(Nat10-ZcKO/WT)]>2). **b** Gene Ontology (GO) enrichment of overlapping genes in **a**. **c-d** Box plots showing the relative mRNA levels of 185 and 265 genes in WT and Nat10-ZcKO oocytes at the GV and MII stages as indicated. Data are presented as mean± SEM, n=3. P-value by two-tailed Student's t-test. **e** The trend patterns of mRNA with (185 genes) or without (265 genes) ac4C modification during the GV-MII transition. Each light-yellow line represents the levels of one RPF, and the blue and red lines represent the median expression levels in WT and Nat10-ZcKO, respectively. **f-g** Box plots showing the relative RPF levels of 185 and 265 genes in WT and Nat10-ZcKO oocytes at the GV and MII stages as indicated. **h** The trend patterns of RPF with (185 genes) or without (265 genes) ac4C modification during the GV-MII transition in WT and Nat10-ZcKO oocytes as in **e**. **i-j** Box plots showing the relative TE levels of 185 and 265 genes in WT and Nat10-ZcKO oocytes at the GV and MII stages as indicated. **k** The trend patterns of TE with (185 genes) or without (265 genes) ac4C modification during the GV-MII transition in WT and Nat10-ZcKO oocytes as in **e**. **l** Heatmap showing the integrative hierarchical clustering for all DEGs with the corresponding RPF levels, TE levels and ac4C levels. The enriched GO terms are listed.

REVIEWERS' COMMENTS

Reviewer #1 (Remarks to the Author):

My concerns have been adequately addressed.

A large amount of new data, including acRIP-seq and Ribo-seq which are technically challenging for mammalian oocytes, have been added during the revision which support the main findings of this study well. This work will be very valuable to the community.

There is one minor point need to be fixed.

The replicates shown in Supplementary Figure 7g are not consistent with the replicates in Supplementary Figure 7d-e. Please correct this. In addition, one of the Y-axis label in Supplementary Figure 7g is wrong as "MT MII".

Reviewer #2 (Remarks to the Author):

Overall, the authors have strengthened the manuscripts with more validation experiments to focus more on the mechanistic role of NAT10 during GV to MII oocyte maturation, which seems to be the main function the gene.

1. Because others had identified and published the important role of Nat10 during oocyte maturation, showing in-depth mechanistic role of Nat10, especially on how it might regulate CCR4-NOT complex, would be noteworthy for publishing a new study on the same topic. In the first version of manuscript, the evidences to support how Nat10 regulated the global translational level during GV to MII maturation were weak. In the revised version, the authors added a few key experiments to address this part. In Figure 10, the authors added Ribo-seq datasets and reported that '...we observed that 491 gene transcripts significantly accumulated in Nat10-ZcKO MII oocytes displayed upregulated RPF enrichment in translome (Fig. 10b, c). GO analysis revealed that those gene transcripts were predominantly enriched in translation, RNA processing and blastocyst formation (Fig. 10d-f), including the gene transcripts from CCR4-NOT complex.'

If the CCR4-complex were also translationally enriched in Nat10-ZcKO MII oocytes, why would the protein level show a significant decrease for these genes? (Figure 8d, Western analysis of CNOT6L, CNOT7 and BTG4). This part of analysis was still confusing. The authors need to better explain how the loss of Nat10 increased the overall transcriptome and translome of a group of genes/proteins such as CCR4-complex but further led to defects in GV to MII maturation.

2. Figure 10j and k were cited to conclude that the overall TE was lower in the mutant, but the analysis was not clearly described especially for 10k to reach the conclusion. What is the criterion for choosing $TE > 3$ as high-TE $TE < 0.33$ as low-TE? This threshold might make the number looks like more genes are low-TE.

3. The protein expression of NAT10 in Figure 8d was too faint or the background was too high to show the difference between WT and the mutant.

4. Was Figure 2e mislabeled? Nat10-ZcKo should be ScKO?

5. Figure 8 and 9 should be moved to supplementary.

Point-by-point letter

Reviewer #1 (Remarks to the Author):

My concerns have been adequately addressed.

A large amount of new data, including acRIP-seq and Ribo-seq which are technically challenging for mammalian oocytes, have been added during the revision which support the main findings of this study well. This work will be very valuable to the community.

There is one minor point need to be fixed.

The replicates shown in Supplementary Figure 7g are not consistent with the replicates in Supplementary Figure 7d-e. Please correct this.

Reply: We have confirmed that these are correct (7g covers all replicates while 7d-f show representative two or three replicates)

In addition, one of the Y-axis label in Supplementary Figure 7g is wrong as "MT MII"

Reply: changed as suggested.

Reviewer #2 (Remarks to the Author):

Overall, the authors have strengthened the manuscripts with more validation experiments to focus more on the mechanistic role of NAT10 during GV to MII oocyte maturation, which seems to be the main function the gene.

1. Because others had identified and published the important role of Nat10 during oocyte maturation, showing in-depth mechanistic role of Nat10, especially on how it might regulate CCR4-NOT complex, would be noteworthy for publishing a new study on the same topic. In the first version of manuscript, the evidences to support how Nat10 regulated the global translational level during GV to MII maturation were weak. In the revised version, the authors added a few key experiments to address this part. In Figure 10, the authors added Ribo-seq datasets and reported that '...we observed that 491 gene transcripts significantly accumulated in Nat10-ZcKO MII oocytes displayed upregulated RPF enrichment in translome (Fig. 10b, c). GO analysis revealed that those gene transcripts were predominantly enriched in translation, RNA processing and blastocyst formation (Fig. 10d-f), including the gene transcripts from CCR4-NOT complex.' If the CCR4-complex were also translationally enriched in Nat10-ZcKO MII oocytes, why would the protein level show a significant decrease for these genes? (Figure 8d, Western analysis of CNOT6L, CNOT7 and BTG4). This part of analysis was still confusing. The authors need to better explain how the loss of Nat10 increased the overall transcriptome and translome of a group of genes/proteins such as CCR4-complex but further led to defects in GV to MII maturation.

Reply: Only few transcripts upregulated from CCR4-NOT complex. For clarification, we have confirmed and removed “including the gene transcripts from CCR4-NOT complex” statement.

2. Figure 10j and k were cited to conclude that the overall TE was lower in the mutant, but the analysis was not clearly described especially for 10k to reach the conclusion. What is the criterion for choosing TE>3 as high-TE TE<0.33 as low-TE? This threshold might make the number look like more genes are low-TE.

Reply: We have tried different cutoff (e.g. TE>2 as high-TE TE<0.5), the conclusion is similar, with the number going down for low-TE.

(Ref: Ultrasensitive Ribo-seq reveals translational landscapes during mammalian oocyte-to-embryo transition and pre-implantation development)

3. The protein expression of NAT10 in Figure 8d was too faint or the background was too high to show the difference between WT and the mutant.

Reply: Nat10 protein itself in MII oocyte is low, which causes high background, but this doesn't impact the conclusion.

4. Was Figure 2e mislabeled? Nat10-ZcKo should be ScKO?

Reply: changed as suggested.

5. Figure 8 and 9 should be moved to supplementary.

Reply: Figure 8 and 9 are relevant to the mechanistic role of Nat10, we thus prefer to keep them as main figures.